



# Contribution and pathways of diazotroph derived nitrogen to zooplankton during the VAHINE mesocosm experiment in the oligotrophic New Caledonia lagoon

**Brian P. V. Hunt[1,2], Sophie Bonnet[3,4], Hugo Berthelot[3], Brandon J. Conroy[5], Rachel A. Foster[6], Marc Pagano[3,4]**

[1] {University of British Columbia, Department of Earth, Ocean and Atmospheric Sciences, Vancouver, V6T 1 Z4, British Columbia, Canada}

[2] {Hakai Institute, P.O. Box 309, Heriot Bay, BC, V0P 1H0, Canada}

[3] {Aix Marseille Université, CNRS/INSU, Université de Toulon, IRD, Mediterranean Institute of Oceanography (MIO) UM 110, 13288, Marseille, France}

[4] {IRD/CNRS/Aix-Marseille University, Mediterranean Institute of Oceanography (MIO) – IRD Noumea, 101 Promenade R. Laroque, BPA5, 98848, Noumea cedex, New Caledonia}

[5] {Department of Biological Sciences, Virginia Institute of Marine Science, College of William and Mary, Gloucester Point, Virginia 23062, USA}

[6] {Department of Ecology, Environment, and Plant Sciences, Stockholm University, Stockholm Sweden 10691}

Corresponding to: Brian Hunt; bhunt@eos.ubc.ca



## 1 Abstract

In oligotrophic tropical and subtropical oceans, where strong stratification can limit the
replenishment of surface nitrate, dinitrogen ($N_2$) fixation by diazotrophs can represent a significant
source of nitrogen (N) for primary production. The VAHINE experiment was designed to examine
the fate of diazotroph derived nitrogen (DDN) in such ecosystems. In austral summer 2013 three
large ($\sim 50$ m$^3$) *in situ* mesocosms were deployed for 23 days in the New Caledonia lagoon, an
ecosystem that typifies the low-nutrient, low-chlorophyll environment, to stimulate diazotroph
production. The zooplankton component of the study aimed to measure the incorporation of DDN
into zooplankton biomass, and assess the role of direct diazotroph grazing by zooplankton as a
DDN uptake pathway. Inside the mesocosms the diatom-diazotroph association (DDA) het-1
predominated during day 5-15 while the unicellular diazotrophic cyanobacteria UCYN-C
predominated during day 15-23. A *Trichodesmium* bloom was observed in the lagoon (outside the
mesocosms) towards the end of the experiment. The zooplankton community was dominated by
copepods (63 % of total abundance) for the duration of the experiment. Using two source N isotope
mixing models we estimated a mean $\sim 30$ % contribution of DDN to zooplankton biomass at the
start of the experiment, indicating that the natural summer peak of $N_2$ fixation in the lagoon was
already contributing significantly to the zooplankton. Stimulation of $N_2$ fixation BNF in the
mesocosms corresponded with a generally low level enhancement of DDN contribution to
zooplankton biomass, but with a peak of $\sim 70$ % in Mesocosm 1 following the UCYN-C bloom.
qPCR analysis targeting four of the common diazotroph groups present in the mesocosms
(*Trichodesmium*, het-1, het-2, UCYN-C) demonstrated that all were ingested by copepod grazers
and that target abundance generally corresponded with their *in situ* abundance. $^{15}N_2$ labeled grazing
experiments provided evidence for direct ingestion and assimilation of UCYN-C-derived N by the
zooplankton, but not for het-1 and *Trichodesmium*, supporting an important role of secondary
pathways of DDN to the zooplankton for the latter groups, i.e., DDN contributions to the dissolved
N pool and uptake by non-diazotrophs. This study appears to provide the first evidence of direct
UCYN-C grazing by zooplankton, and indicates that UCYN-C-derived N contributes significantly
to the zooplankton food web in the New Caledonia lagoon though a combination of direct grazing
and secondary pathways.



## 1 Introduction

Dinitrogen ($N_2$) fixation by diazotrophs is considered to be the most important external source of reduced nitrogen (N) for the ocean, exceeding atmospheric and riverine inputs (Gruber et al., 2004). The nitrogenase enzyme gives diazotrophs the capacity to reduce $N_2$ gas into bioavailable ammonium. This new N is particularly important in the oligotrophic tropical and subtropical oceans, where strong stratification limits the upward mixing of nitrate replete deep water into the photic zone, sustaining ~50 % of primary productivity (Karl et al., 1997). In addition, some experimental research indicates that $N_2$ fixation will be enhanced by rising atmospheric carbon dioxide ($CO_2$) concentrations and ocean warming, highlighting a potentially increasingly important role of diazotrophs in the oceanic carbon and N cycles (Hutchins et al., 2009; Hutchins et al., 2007; Levitan et al., 2007; Sheridan and Landry, 2004).

Stable isotope analysis has served as a powerful tool for investigating the contribution of new N to pelagic food webs (Carpenter et al., 1999; Hannides et al., 2009; Landrum et al., 2011; Mompean et al., 2013; Montoya et al., 2002). $N_2$ gas has an N isotope ratio ($\delta^{15}N$) of 0 ‰ and preferential uptake of $^{14}N$ leads to $\delta^{15}N$ as low as -2.5 ‰ for diazotrophs (Montoya et al., 2002). By comparison, the average ocean nitrate $\delta^{15}N$ is ~ 5 ‰ (Sigman et al., 1999; Sigman et al., 1997), leading to higher $\delta^{15}N$ for primary producers using this source. The $\delta^{15}N$ signatures of zooplankton reflect the balance between these contrasting N sources, the relative contributions of which can be estimated using a two part mixing model (Montoya et al., 2002). This modeling approach has been used to demonstrate a significant contribution of diazotroph derived N (DDN) to particulate matter and zooplankton biomass (Aberle et al., 2010; Landrum et al., 2011; Loick-Wilde et al., 2012; Mompean et al., 2013; Montoya et al., 2002; Sommer et al., 2006; Wannicke et al., 2013), and transfer of DDN beyond zooplankton to micronekton (Hunt et al., 2015). However, despite this measured contribution of DDN, questions remain as to the pathways of DDN into marine food webs (Wannicke et al., 2013).

Cyanobacteria are considered the major $N_2$-fixing microorganisms in the ocean (Zehr, 2011). The open ocean diazotrophic cyanobacteria can be divided into three groups (Luo et al., 2012): (1) non-heterocystous filamentous cyanobacteria, e.g. *Trichodesmium* spp. (Capone et al., 2005); (2) heterocystous cyanobacteria frequently found in association with diatoms (diatom-diazotroph associations (DDAs; see review by (Foster and O'Mullan, 2008)), e.g., *Richelia* in association with *Rhizosolenia* and *Hemiaulus* (*Rhizosolenia* and *Hemiaulus* are often referred to and quantified by





the *Richelia* strain that associates with each, het-1 and het-2, respectively); and (3) unicellular
cyanobacterial lineages (UCYN-A, B, and C), with a size range of between 1 and 6 μm (Moisander
et al., 2010). Until recently research related to the role of fixed N in marine food webs has largely
focussed on *Trichodesmium* spp. It is generally considered that the majority of *Trichodesmium*
DDN reaches the food web through the release of dissolved N (Capone et al., 1994; Glibert and
Bronk, 1994; Mulholland and Bronk, 2004; Mulholland and Capone, 2001) which is taken up by
heterotrophic and autotrophic microbes (Bonnet et al., in revision), and which are subsequently
consumed by the zooplankton (Capone et al., 1997; O'Neil and Roman, 1992). Dissolved N is
released through a combination of endogenous and exogenous processes, including viral lysis
(Hewson et al., 2004), zooplankton sloppy feeding (O'Neil et al., 1996), or programmed cell death
(Berman-Frank et al., 2004). Recent research has demonstrated that UCYN can release similar
amounts of dissolved N to *Trichodesmium* (Berthelot et al., 2015a).
The direct pathway of DDN to pelagic food webs, via zooplankton grazing, has been considered
limited due to cyanobacteria possessing cyanotoxins (Guo and Tester, 1994), large cell size in the
case of filamentous cyanobacteria such as *Trichodesmium* spp. and *Nodularia* spp. and poor
nutritional quality (O'Neil and Roman, 1992; O'Neil, 1999). Experimental studies of direct
zooplankton grazing on cyanobacteria have yielded conflicting results. Reduced feeding and egg
production rates were measured for the Baltic Sea calanoid copepods *Eurytemora affinis* and
*Acartia bifilosa* when fed a mixed cyanobacteria diet, while others (Koski et al., 2002) reported
that *A. bifilosa* feeding and egg production rates were unaffected by a diet of *Nodularia* spp.. In
another Baltic Sea study, direct grazing of cyanobacteria was demonstrated to be more prevalent
amongst cladocera (small crustacean) than copepods, and that they favoured the cyanobacterium
*Aphanizomenon* over *Nodularia* (Wannicke et al., 2013). Direct grazing on *Trichodesmium* spp.
has been demonstrated for the harpacticoid copepod *Macrosetella gracilis*, *Miracia efferata,* and
*Oculosetella gracilis* in the Caribbean (O'Neil et al., 1996; O'Neil and Roman, 1994) and *Acartia*
*tonsa* in the north Atlantic (Guo and Tester, 1994). In the north Atlantic, stable isotope measured
zooplankton DDN uptake suggested enhanced uptake when DDA abundance was higher than
*Trichodesmium* spp., though the actual DDN uptake pathways could not be determined (Montoya
et al., 2002). Combined, the results of previous research indicate that direct grazing can be an
important pathway of DDN into marine food webs, but that it is dependent on both the
cyanobacteria and zooplankton community composition.



The New Caledonian coral lagoon in the southwestern Pacific is a tropical low-nutrient low-
chlorophyll (LNLC) system. Oligotrophic ocean water enters the lagoon from the south and is
driven north by the trade winds and tidal forcing before exiting through several deep inlets in the
intertidal barrier reef that forms the western boundary of the lagoon (Ouillon et al., 2010). Primary
productivity is N-limited throughout the year (Torréton et al., 2010), giving $N_2$-fixing
microorganisms a competitive advantage over non-diazotrophic organisms. High abundance of
diazotrophs have been reported during the austral summer, for both *Trichodesmium* spp. (Rodier
and Le Borgne, 2010) and UCYN (Biegala and Raimbault, 2008). The New Caledonian lagoon
therefore represents an ideal location to investigate the ecosystem role of diazotrophs.
Accordingly, this location was selected for the implementation of the 23 day VAHINE mesocosm
experiment in the austral summer of 2013. A full description of this experiment is provided by
Bonnet et al. (2015), with core details outlined in the methods below. VAHINE was designed
specifically to investigate the fate of DDN in the ecosystem, i.e., its transfer to the planktonic food
web and its contribution to export production (Bonnet et al., in preparation). Here we present the
zooplankton component of the VAHINE program. Our aims were 1) to measure the contribution
of DDN to zooplankton biomass, and 2) investigate the role of direct grazing by zooplankton on
diazotrophs as a pathway for DDN into the zooplankton food web.
## 2    Material and methods
### 2.1.  Mesocosms description and zooplankton sampling and processing
Briefly, during VAHINE three large volume (~50 m³) mesocosms (M1-3) were deployed 28 km
off the coast (22° 9.10 S; 166° 26.90 E) in the south-west (Noumea) of the New Caledonian lagoon,
from 13 January 2013 (day 1) to 4 February 2013 (day 23). The site was located at a depth of 25
m, in close proximity to Boulari passage and thus strongly influenced by oceanic oligotrophic
waters coming from outside the lagoon.  Each mesocosm enclosure comprised a cylindrical bag
2.3 m in diameter and 15 m deep. The mesocosms open tops were maintained at a height of ~1 m
above the surface to prevent external water additions. Screw-top plastic bottles (250 mL) were
attached to the bottom of the mesocosms to collect sinking particles, and these were serviced daily
by scuba divers. To alleviate potential phosphorus limitation and intentionally stimulate
diazotrophy, the mesocosms were fertilized with ~0.8 µmol $L^{-1}$ of dissolved inorganic phosphorus
(DIP) on day 4 of the experiment. Physical conditions (Bonnet et al., 2015), primary production





and $N_2$ fixation rates (Berthelot et al., 2015b) were monitored daily in the mesocosms and in an adjacent control site throughout the experiment (hereafter called lagoon waters), the methods and results of which are described in detail in the cited publications.

Zooplankton were sampled on seven occasions from the three mesocosms and lagoon waters (the control site), at intervals of every 3 to 4 days, always between 9:30 and 10:30 am. Sampling was with a 30 cm diameter, 100 cm long, 80 µm mesh net fitted with a filtering cod end. On each sampling occasion three vertical hauls (hereafter called Samples 1, 2 and 3) were collected from the upper 10 m of each site. The total volume sampled on each occasion (sum of the three nets) was 2.13 $m^3$, representing 4 % of the total mesocosm volume. As reported below, zooplankton densities did not vary appreciably over the course of the experiment, indicating that the sampling did not significantly impact the mesocosm communities.

All zooplankton samples were stored in a cooler and returned to the Amedee Island field station located 1 nautical mile from the mesocosms site for processing within 30-60 minutes of the final net haul. Zooplankton Sample 1 was split in half and one half preserved in 4 % buffered formaldehyde for community composition analysis and the other half filtered onto a pre-combusted 25 mm GF/F filter for measurement of total zooplankton biomass. Sample 2 was filtered onto a pre-combusted (450°C, 4 h) 25 mm GF/F filter for stable isotope analysis. Sample 3 was drained using a 64 µm sieve within 60-90 minutes of collection, and held in its original collection jar in an insulated cool container with ice packs until returning to the Noumea laboratory for processing ~ 6 h later. In the Noumea laboratory, Sample 3 was filtered onto a 2 µm polycarbonate filter and then frozen in a cryovial at -80°C for molecular analysis of zooplankton gut contents.

Taxonomic analysis of the zooplankton community was completed using a stereo microscope, from a 1/8 to 1/16 fraction of each sample. Specimens were identified to the level of order and enumerated. The category copepod nauplii comprised a mix of calanoid, cyclopoid and poecilostomatoid copepods. No flowmeter was used with the nets and counts were converted to individuals $m^{-3}$ assuming that the net sampled with 100 % efficiency. Samples for biomass estimation were rinsed with ammonium formate to remove salt, dried at 50°C for 48 h, and weighed to the nearest 0.01 mg using a microbalance. Values were converted to mg Dry Weight (DW) $m^{-3}$.



Zooplankton samples for stable isotope analysis were first dried at 50°C for 48 h. Zooplankton
were subsequently removed from the GF/F filter, homogenized using a mortar and pestle, and
packaged into ~ 1 mg sub-samples. Stable isotope analysis of these samples was performed at the
IsoEnvironmental Laboratory (http://www.isoenviron.co.za/), Rhodes University, South Africa,
with a Europa Scientific 20-20 isotope ratio mass spectrometer (IRMS) linked to a preparation unit
(ANCA SL). Casein and a mixture of beet sugar and ammonium sulphate were used as internal
standards and were calibrated against the International Atomic Energy Agency (IAEA) standards
CH-6 and N-1) and the IRMS certified reference material EMA-P2 (see Certificate BN/132357).
$\delta^{13}C$ and $\delta^{15}N$ were determined in parts per thousand (‰) relative to external standards of Vienna
Pee Dee Belemnite and atmospheric N. Repeated measurements of an internal standard indicated
measurement precision of ±0.09 ‰ and ±0.19 ‰ for $\delta^{13}C$ and $\delta^{15}N$ respectively.
The $\delta^{15}N$ of Suspended Particulate Matter ($PN_{susp}$) was measured daily in each mesocosm and in
lagoon waters to provide a baseline value for the pool of particles available for zooplankton
grazing. Discrete water samples were collected daily from 6 m depth and filtered onto pre-
combusted 25 mm GF/F filters. $\delta^{15}N$ values were determined by high-temperature combustion
coupled with isotope ratio mass spectrometry using a Delta Plus Thermo Fisher Scientific mass
spectrometer (Knapp et al., in preparation).

**2.2.  Zooplankton DNA extraction and quantitative PCR (qPCR)**

Individual copepods were picked from each filter and identified to order (Calanoid, Harpactacoid,
or Cyclopoid). Copepods were then placed in autoclaved artificial seawater (ASW) and visually
inspected under a dissecting microscope for contamination from phytoplankton and detritus
particularly in the mouthparts and appendages. Large particles were picked clean from the
mouthparts and appendages with 20µm minutien pins (Fine Science Tools, Foster City, CA USA)
before subsequently rinsing through 5 sterile baths of autoclaved ASW water and a final inspection
under an epifluorescence microscope equipped with blue (450-490 nm) and green (510-560 nm)
excitation filters (Boling et al., 2012). Number of copepods and composition varied with each tow
and can be found in Table 1. Aside from the day 5 samples from M1, where copepods were
extracted by order, all copepods per sample were pooled together for extraction. DNA extraction
was performed with the Qiagen DNeasy® Blood and Tissue Kit using slight modifications to the
manufacturers "Animal Tissue (Spin-Column)" protocol. An overnight (12 hour) lysis step was



performed, all reagent volumes were 50 % of the manufacturer's suggestions, and the final elution
volume was 35 µl in the provided "Buffer AE."
For the qPCR assays, we used the TaqMAN primers and probes described by (Church et al., 2005)
for *Trichodesmium* spp.*,* het-1 (*Richelia* associated with the diatom *Rhizosolenia*) and het-2
(*Richelia* associated with the diatom *Hemiaulus*), and unicellular group C (UCYN-C) primers and
probes described by (Foster et al., 2007). The 4 target diazotrophs were selected based on their
being the most abundant $N_2$ fixers throughout the mesocosm experiment (Turk-Kubo et al., 2015).
For all TaqMAN PCR, the 20 µL reactions contained 10 µL of 2X Fast Advanced Master Mix
(Applied Biosystems, Stockholm Sweden), 5.5 µL of nuclease free water, 1.0 µL each of Forward
and Reverse Primer (0.5 µmol $L^{-1}$) and 0.5 µL of fluorogenic probe (0.25 µmol L-1) and 2 µL of
template. Each reaction was performed in triplicate and 2 µL of no template controls (NTCs) were
run.   All PCR amplifications were conducted in an ABI Step One Plus system (Applied
Biosystems) with the following parameters: 50 ºC for 2 min., 95 ºC for 20 s, and 40 cycles of 95
ºC for 1 s, followed by 60 ºC for 20 s.   Gene copy abundances were calculated from the mean
number of cycle ($C_t$) of the three replicates and the standard curve for the appropriate primer and
probe set (see below). In samples where one or two of the three replicates produced an
amplification signal, these are noted as detectable but not quantifiable.
For each primer and probe set, duplicate standard curves were made from 10-fold dilutions ranging
from 1 to $10^8$ copies per reaction. The standards curves were synthesized 359 bp gene fragments
(gBlocks, Integrated DNA Technologies, Leuven, Beligium) of the nifH gene. Regression analyses
of the number of cycles ($C_t$) of the standard curves were calculated in Excel.

### 23   2.3. Zoopankton ingestion of diazotrophs: $^{15}N_2$ labeled grazing experiments

Direct grazing by zooplankton on diazotrophs was assessed by a series of three $^{15}N_2$ labeling
experiments. Each experiment consisted of $^{15}N_2$-labeled bottle incubations of freshly collected
zooplankton in the presence of natural phytoplankton assemblages. The $^{15}N_2$ label was taken up by
the diazotroph in the incubation bottle and used as a marker of zooplankton diazotroph ingestion.
For each experiment (E1, E2 and E4), zooplankton was collected after sunset (18:00-19:00 h) by
repeated 1m $s^{-1}$ vertical hauls with the same net used for daytime zooplankton collections (see
above), in close proximity to the mesocosms site. Live zooplankton were collected with a 64 µm
sieve and placed in three 25 L polycarbonate carboys (two net tows per carboy) filled with seawater



collected using a Teflon pump (St-Gobain Performance Plastics) from M1 (1 m depth) on day 12
for experiment E1, during a DDA dominated period (> 80 % of diazotroph community comprised
*Richelia* associated with *Rhizosolenia*, i.e., het-1); from M2 (1 m depth) on day 17 for experiment
E2, during a UCYN-C bloom (comprising > 80 % of diazotroph community); and from lagoon
waters (1 m depth) on day 23 for E4 during a *Trichodesmium* spp. bloom (comprising > 80 % of
diazotroph community) (Turk-Kubo et al., 2015). Although each experiment was > 80 %
dominated by a single diazotroph species, it must be noted that each contained other diazotroph
species. Carboys were filled to the top, leaving no head space, and tightly closed with septum caps.
Carboys were immediately amended with 26 ml $^{15}N_2$ gas (Cambridge isotopes, 98.9 atom% $^{15}$N)
using a gas-tight syringe, gently agitated 20 times to facilitate the $^{15}N_2$ bubble dissolution, and
incubated in situ on a mooring line close to the mesocosms site at the sampling depth (1 m) for 24
to 96 h.
Zooplankton T0 atomic enrichment was measured in triplicate for E1 and the average value was
used as the baseline for E1, E2 and E4. Incubation termination times were 24, 48, and 72 h for E1;
24, 72, 96 h for E2; 24 and 40 h for E4 (Table 2). After incubation, animals were recovered from
each carboy by gravity filtration onto a 64 µm mesh sieve, transferred to a 20 µm polycarbonate
filter and frozen until the end of the VAHINE experiment. Subsequently, the zooplankton on the
filters were identified to order and enumerated under a stereo microscope (Table 2) before being
dried at 24 h at 60 °C. In all cases composition comprised an 87-100 % mix of Poecilostomatoid
and Calanoid copepods. All individuals from each time point were pooled for measurement of bulk
zooplankton PON $^{15}$N enrichment, using a Delta plus Thermo Fisher Scientific isotope ratio mass
spectrometer (Bremen, Germany) coupled with an elemental analyzer (Flash EA, ThermoFisher
Scientific).
The atomic enrichment of the dominant diazotrophs during each experiment were measured after
24 hour incubation in a parallel experiment, using the same enrichment procedure as the
zooplankton grazing experiment, designed to trace the fate of DDN in phytoplankton (Berthelot et
al., 2015b; Bonnet et al., in revision; Bonnet et al., in review). Accordingly, atomic enrichment
was obtained for UCYN-C (E2) and *Trichodesmium* spp. (E4), but not for DDA (E1).
**2.4.  Statistical analyses**



A sample by taxon matrix was created using taxon specific densities. Densities were fourth root
transformed and the percentage similarity between stations from all surveys was calculated using
the Bray-Curtis similarity index (Field et al., 1982). The similarity matrix was then ordinated using
non-metric multidimensional scaling (NMDS), summarising between sample variation in
community composition into two dimensions.  This multivariate analyses were performed using
PRIMER 6 (Clarke and Warwick, 2001). The NMDS had a stress value of 0.23. The first two
dimensions of the ordination were plotted against sampling date for each mesocosm and the lagoon
site to enable visual assessment of the change in zooplankton composition over the course of the
experiment.
**2.5.  Calculation of DDN contribution to zooplankton biomass**
The contribution of DDN (%) to zooplankton $\delta^{15}$N (ZDDN) in each sample collected during this
study was calculated using a two source mixing model following (Sommer et al., 2006):
Equation 1:    $\% \, ZDDN = 100 * \left( \dfrac{\delta15N_{zpl} - \delta15N_{zplref}}{TEF + \delta15N_{diazo} - \delta15N_{zplref}} \right)$
where $\delta^{15}$N$_{zpl}$ is the isotopic signature of the zooplankton collected during the experiment; TEF is
the trophic enrichment factor, which was set at 2.2 (McCutchan et al., 2003; Vanderklift and
Ponsard, 2003); $\delta^{15}$N$_{diazo}$ is the isotopic signature of diazotrophs, set at -2 ‰ (Montoya et al.,
2002); $\delta^{15}$N$_{zplref}$ is the isotopic signature of zooplankton assuming nitrate based phytoplankton
production, set at 6.7 ‰ assuming a baseline nitrate $\delta^{15}$N of 4.5 ‰ (Montoya et al., 2002) and a
TEF of 2.2. Daily DDN production ingested by the zooplankton each day (mg Dry Weight day$^{-1}$)
was calculated as follows:
Equation 2:   daily DDN ingested day$^{-1} = \left( \dfrac{N \, production + N \, excretion}{assimilation \, efficiency} \right) * \% \, ZDDN$
Calculations were based on production and excretion values measured by (Le Borgne et al., 1997)
in Uvea Lagoon. These authors measured rates for two size classes (35-200 μm and 200–2000
μm). Since our sampling spanned both of these size classes we used mean rates: daily zooplankton
production (mg DW d$^{-1}$) was calculated using a Production: Biomass ratio of 114 %; daily





excretion assuming a net growth efficiency (K) of 0.513; ingestion assuming an assimilation
efficiency of 0.7; and N content (mg DW) using the value of 4.25 % for a mixed zooplankton
community. Finally, we estimated the percentage of daily DDN production consumed by
zooplankton:
Equation 3:
$$\% \text{ daily DDN production ingested day}^{-1} = 100 * \left( \frac{\text{daily DDN ingested day} - 1}{\text{daily DDN production}} \right)$$
Daily DDN production (N$_2$ fixation) was calculated from the mean of the three measurement
depths in each mesocosms (Berthelot et al., 2015b).

## 3  Results

### 3.1. Environmental context

Briefly, seawater temperature increased inside the mesocosms and in Noumea lagoon waters from
25.5 to 26.2 ºC over the course of the experiment. The water column was well mixed in the
mesocosms as temperature and salinity were homogeneous with depth over the course of the
experiment (Bonnet et al., in preparation). Prior to the DIP fertilization on day 4 (hereafter called
P0), DIP concentrations in the mesocosms ranged from 0.02 to 0.05 µmol L$^{-1}$ (Berthelot et al.,
2015b). The day after the fertilization, DIP concentrations were ~ 0.8 µmol L$^{-1}$ in all mesocosms.
Subsequently the concentrations decreased steadily towards initial concentrations by the end of
the experiment. Depth averaged nitrate+nitrite concentrations were below 0.04 µmol L$^{-1}$ the day
before DIP fertilization and decreased to 0.01 µmolL$^{-1}$ towards the end of the experiment. In
lagoon waters, nitrate+nitrite remained below 0.20 µmolL$^{-1}$ and DIP averaged 0.05 µmolL$^{-1}$
throughout the experiment.
Bulk N$_2$ fixation rates averaged 18.5±1.1 nmol N L$^{-1}$ d$^{-1}$ over the 23 days of the experiment in the
three mesocosms (all depths averaged together; (Bonnet et al., in review)).  Rates increased
significantly in the mesocosms over the course of the experiment to reach an average of 27.3±1.0
nmol N L$^{-1}$ d$^{-1}$ during the second half of the experiment (day 15 to day 23, hereafter called P2)
(Bonnet et al., in review). N$_2$ fixation rates measured in the lagoon waters were significantly




(p<0.05) lower than those measured in lagoon waters (9.2±4.7 nmol N L$^{-1}$ d$^{-1}$) over the 23 days of
the experiment. They did not differ significantly over the experimental period.
The diazotroph assemblage in the lagoon on the day that the mesocosm experiment was initiated
was composed primarily of DDAs (het-1: *Richelia* associated with *Rhizosolenia*; and het-2:
*Richelia* associated with *Hemiaulus*) and the symbiotic UCYN-A2 and A1 (Turk-Kubo et al.,
2015). *Trichodesmium* spp. and UCYN-C were minor components, and at least an additional three
phylotypes were present, including one heterotrophic diazotroph. The abundance and community
of diazotrophs changed extensively in the mesocosms over the course of the experiment. From day
1 to 4 a shift in the starting community was observed in the mesocosms. Het-1 remained the most
abundant diazotroph, however, UCYN-A2 abundances decreased and *Trichodesmium* spp.
abundances increased with respect to their abundances in the lagoon, while UCYN-C remained at
low abundance levels. After DIP fertilisation, from day 5 to day 14 (hereafter called P1), the
abundance of het-1 increased. Following day 15 the community shifted towards dominance of
UCYN-C, the abundance of which increased substantially during P2 (Turk-Kubo et al., 2015).
Het-1 was the dominant diazotroph in the lagoon waters where a *Trichodesmium* spp. bloom began
to develop during P2, after day 20 (Turk-Kubo et al., 2015). Chlorophyll a (Chl a) biomass was <
0.3 µg L$^{-1}$ in all three mesocosms during P0 and P1 (Leblanc et al., in preparation). During P2, Chl
a increased in all the mesocosms, but particularly M3, reaching maximum depth-averaged
concentrations of 0.55, 0.47 and 1.29 µg L$^{-1}$ in M1, M2 and M3, respectively. Lagoon Chl a
followed a similar pattern to the mesocosms, being < 0.3 µg L$^{-1}$ during the P0 and P1 timeframe,
and increasing to a lower extent to 0.42 µg L$^{-1}$ during P2.

### 3.2. Zooplankton

Zooplankton abundance at the start of the experiment averaged ~ 5,000 ind m$^{-3}$ in lagoon waters,
M1 and M2, while it was 10,735 ind m$^{-3}$ in M3 (Figure 1). Over the course of the experiment
abundance in M1 and M2 ranged between 5425 and 1741 ind m$^{-3}$. M1 densities had a slight
declining trend, while M2 densities were relatively stable, even increasing towards the end of the
experiment. In M3, zooplankton abundance was consistently higher than M1 and M2 though
declining after day 12 from 6618 ind m$^{-3}$ to 4256 ind m$^{-3}$ on day 23. The lagoon waters differed
from the mesocosms with zooplankton abundance levels increasing to peak at 13,113 ind m$^{-3}$ on
day 16, before declining to ~ 7,300 ind m$^{-3}$ on day 23. Zooplankton had a mean biomass of 24 mg



DW m$^{-3}$ and ranged between 17.2 and 40 mg DW m$^{-3}$ (Figure 1). No consistent temporal pattern
in zooplankton biomass was detected over the course of the experiment.
The zooplankton community was dominated by copepod nauplii at all sites, with the exception of
day 2 at M2 when poecilostomatoids dominated and day 9 at M1 when appendicularians
dominated (Figure 2). Copepod nauplii contributed an average of 51 % to total abundance (2784
ind m$^{-3}$). Appendicularians were the next most abundant group, contributing an average of 15.1%
to total abundance (801 ind m$^{-3}$), followed by poecilostomatoid copepods at 11.5 % (541 ind m$^{-3}$).
Peaks in appendicularian abundance were observed during P1 in M1 and M3. Cyclopoid, calanoid
and harpacticoid copepods contributed 5.5, 5, and 1.4 % to total abundance respectively. Although
the proportional contributions of these groups was low, their abundance levels were relatively high,
averaging 276, 265, and 72 ind m$^{-3}$ for cyclopoid, calanoid and harpacticoids, respectively.
Bray Curtis similarity levels among samples exceeded 70 % in all cases with the exception of the
day 19 control sample (~ 65 %). This is on the high range of similarity for zooplankton
communities (Hunt et al., 2008). The first dimension of the NMDS was most variable over the
course of the experiment, and between site variability was highest on day 2 (Figure 3). Subsequent
to day 2, NMDS scores for the three mesocosm converged, with M1 and M2 having the greatest
similarity. The NMDS scores for Dimension 1 in all mesocosms diverged from the lagoon waters
after day 9. The opposite directional trends of the mesocosms versus the lagoon waters was driven
primarily by changes in abundance levels of the same pool of species.
Zooplankton $\delta^{15}$N averaged 4.9, 4.2, 4.8 and 5.2 ‰ in lagoon waters, M1, M2, and M3,
respectively (Figure 4). Zooplankton $\delta^{15}$N were relatively consistent over the course of the
experiment in M2 and M3. In M1, zooplankton $\delta^{15}$N decreased from a mean of 5 ‰ between day
2 and 12 (P0 and P1) to a mean of 3.2 ‰ from day 16 to 23 (P2). In lagoon waters, a decline in
zooplankton $\delta^{15}$N was evident over the course of the experiment, from 6.02 ‰ on day 5 to 4.38 ‰
on day 23.
The $\delta^{15}$N of PN$_{susp}$ was more variable than the zooplankton, commensurate with the expected
higher cellular turnover rates of the PN$_{susp}$ constituents relative to zooplankton. In M3, PN$_{susp}$ $\delta^{15}$N
increased to the same level as the zooplankton on day 11 and remained at that level until the end
of the experiment. An increase in PN$_{susp}$ $\delta^{15}$N to above zooplankton levels was observed in lagoon
waters and M2 after day 20. Zooplankton $\delta^{15}$N averaged 1.2 ‰ higher than PN$_{susp}$ across all sites,
less than the expected 2.2 ‰ one trophic level difference between the PN$_{susp}$ and zooplankton.





The percent contribution of DDN to zooplankton biomass averaged 30 % (range = 15 to 70 %) in the mesocosms and 28 % (range = 11 to 38 %) in the lagoon waters (Figure 5) over the 23 days experiment. The highest percent contribution of DDN to zooplankton was measured in M1 on day 16 (70 %). The contribution of DDN to zooplankton biomass in M2 and the lagoon increased steadily from ~ 20 % in the middle of P1 (day 9) to 38 % by the end of the experiment. An initial increase in the contribution of DDN to zooplankton biomass was observed in M1 and M3 after 9 until day 16, after which it declined until the end of the experiment despite these mesocosms having the highest $N_2$ fixation rates (Bonnet et al., in review).

Estimated daily DDN production ingested by the zooplankton reached > 100 % across all conditions between day 2 and 9, but decreased in both the mesocosms and lagoon waters after day 9. The decrease was greatest in the mesocosms, corresponding with the higher $N_2$ fixation rates in these sites (Bonnet et al., in review). By the end of the experiment, daily DDN production ingested was 22-34 % across the three mesocosms. In lagoon waters, where $N_2$ fixation rates were lower, daily DDN production ingested ranged between 111 and 61 % until day 23.

### 3.3. Quantitative PCR (qPCR)

In general, the qPCR was successful in amplifying and detecting the 4 different targets (het-1, het-2, *Trichodesmium* spp., and UCYN-C) in the copepods collected during the mesocosm experiment. Poor detection was listed as either below detection (bd) or detectable but not quantifiable (dnq) (see methods).

Of all the oligonucleotides tested, the het-2 and *Trichodesmium* spp. targets were the least detected. However when het-2 and *Trichodesmium* spp. targets were detected, the abundance was high, e.g., 62.1 and 264.4 *nifH* copies/copepod respectively, in M2 during P0 (day 2). Subsequently het-2 detection was bd for the remainder of the experiment, with the exception of two dnq samples, one from the lagoon during P0 (day 2) and another from M2 towards the end of P1 (day 12). *Trichodesmium* spp. targets were bd after day 2, until 277.9 *nifH* copies/copepod was quantified from a M2 sample on day 16. Overall, *Trichodesmium* spp. was more prevalent during P2, being quantifiable or dnq in 5 of 9 samples. Het-1 and UCYN-C were higher in detection, each being bd in only 6 of the 19 samples tested. Het-1 targets were the most frequently detected, occurring at high abundance (16.5-173.3 *nifH* copies/copepod) in all of the mesocosms and lagoon waters during P1 and the beginning of P2, but were bd or dnq after day 19. UCYN-C was detected most





frequently and at highest abundance during P2, corresponding with this groups peak occurrence in
the mescosms.

### 3.4. $^{15}N_2$ labeled grazing experiments on zooplankton

After 24 h incubation the atomic enrichment of UCYN-C was 1.515 atom % and *Trichodesmium*
spp. 0.613 atom %. No direct measurement of atomic enrichment was obtained from DDA. The
average atomic enrichment of zooplankton at T=0 in E1 was 0.373±0.005 atom %. This T0 value
was applied as the baseline for E2 and E4. Zooplankton showed weak atomic enrichment over the
course of E1 (het-1 dominated diazotroph community) and none over the course of E4
(*Trichodesmium* spp. dominated diazotroph community) (Figure 6). Conversely, a large increase
of ~ 0.1 atom% was measured over the course of E2 (UCYN-C dominated diazotroph community).
Although E1 and E4 were of shorter duration than E2, discernable atomic enrichment was
measured in E2 even after 24 h. The only instance where the dominant diazotroph in the water
collected on the day of experiment initiation was also detected in high abundance in copepod guts
on or within one day of this water collection was E2 / UCYN-C (Table 1; Figure 6). *Trichodesmium*
spp. was dnq in copepod guts on day 23 in the lagoon (E4), while there was no evidence of het-1
in copepod guts on day 12 (E2).

## 4   Discussion

The zooplankton biomass sampled during VAHINE, both inside the mesocosms and in lagoon
waters, was is the normal range for the New Caledonian lagoon (Le Borgne et al., 2010). Over the
course of the experiment ~ 28 % of the total volume of each mesocosm was sampled. An additional
2-5 % of the zooplankton community was lost to the mesocosm sediment traps and qualified as
swimmers (Berthelot et al., 2015b). These two sources of losses likely accounted for the slight
declining trend in abundance in M1 and M2, and M3 after day 12. Despite the divergence of lagoon
waters and mesocosms abundance levels over the course of the experiment, a high level of
similarity (> 70 %) was maintained in the community composition among sites, indicating that the
mesocosm zooplankton communities remained largely representative of the natural lagoon
conditions. On average this community comprised 63 % copepods, with the next highest
community contributor being appendicularians (~ 15%). Harpacticoid copepods, which have
previously been noted as important diazotroph grazers contributed < 1.5 % on average.





The $\delta^{15}$N of $PN_{susp}$ over the course of the experiment was high in comparison to measurements
from other areas of the world's oceans with significant $N_2$ fixation (Altabet, 1988; Dore et al.,
2002; Montoya et al., 2002). It has been noted that elevated $\delta^{15}$N of $PN_{susp}$ in the New Caledonian
lagoon may be influenced by island runoff, and particularly untreated sewage which typically has
a $\delta^{15}$N of 5‰ to 20‰ (Cole et al., 2004). Although the VAHINE site was located 28 km from the
coast, and strongly influenced by inflowing oceanic water, the elevated $\delta^{15}$N of $PN_{susp}$, despite a
high contribution of $N_2$ fixation, indicated that the $\delta^{15}$N of $PN_{susp}$ was influenced by land-derived
inputs (Knapp et al., in preparation). Notably the $\delta^{15}$N of $PN_{susp}$ did not show a decreasing trend
over the course of the experiment, either inside or outside the mesocosms, even increasing in M3
during P2, despite the increasing $N_2$ fixation rates in all mesocosms. In contrast, the $\delta^{15}$N of $PN_{susp}$
settling in the sediment traps decreased with time from 4.2±0.2 ‰ during P0, to 3.0±0.4 ‰ during
P1 and 2.3±0.9 ‰ during P2 (Knapp et al., in preparation). Indeed, it is estimated that the majority
of the DDN that accumulated over the course of the experiment was exported to the sediment traps,
either through direct sedimentation of diazotrophs or of non-diazotrophic phytoplankton that had
taken up dissolved N sourced from the DDN pool (Bonnet et al., in review).
Overall, zooplankton $\delta^{15}$N in the mesocosms and lagoon tended to decline gradually over the
course of the experiment, with the exception of M1 where a more marked decline was observed
during P2. A similar, albeit shorter (9 days), mesocosm study conducted in the Baltic Sea measured
a rapid decrease in zooplankton $\delta^{15}$N in response to a *Nodularia spumigena* bloom (Sommer et al.,
2006). In that study elevated zooplankton $\delta^{15}$N (9.9 ‰) at the start of the experiment likely
amplified the effect of DDN uptake. During VAHINE, zooplankton $\delta^{15}$N was ~ 5 ‰ at the start of
the experiment, and the estimated mean contribution of DDN to zooplankton biomass on day 2
was ~ 30 %. As previously mentioned, diazotroph activity in the New Caledonian lagoon peaks in
the summer months (Biegala and Raimbault, 2008; Le Borgne et al., 2010). A time series of
monthly zooplankton samples collected between October 2012 and July 2014 reveals a seasonal
summer depletion of $\delta^{15}$N in the New Caledonia lagoon (B. Hunt, unpublished data). It is therefore
not surprising that a depletion in zooplankton $\delta^{15}$N was less marked during VAHINE, which took
place during the summer season, despite the increase in $N_2$ fixation rates observed at all sites
through the experiment.
The gradual decline of zooplankton $\delta^{15}$N corresponded with the increased contribution of DDN to
zooplankton biomass over the course of the experiment in both the mesocosms and lagoon, with





the exception of M3. The peak DDN contribution to the zooplankton of 70 %, on day 16 in M1,
was on the high end of values reported in the literature (subtropical north Atlantic (Landrum et al.,
2011). The DDN contribution to the zooplankton (~ 30 %) was within the range of estimates for
the subtropical north Atlantic (Landrum et al., 2011; Mompean et al., 2013; Montoya et al., 2002),
Baltic Sea (Sommer et al., 2006; Wannicke et al., 2013), and pelagic waters off the New
Caledonian shelf (Hunt et al., 2015). The gradual decline of zooplankton $\delta^{15}N$ did not match the
large increase in $N_2$ fixation rates measured during VAHINE, evident in the declining percent
DDN ingested.day$^{-1}$, particularly during P2. This may be explained in part by a lag between
ingestion and assimilation of DDN (Rolff, 2000). However, the primary factor was most likely the
rapid export of DDN from the water column limiting zooplankton ingestion of new DDN
production (Bonnet et al., in review).
The combination of qPCR and $^{15}N_2$ labeled grazing experiments provided insights into the
potential role of direct grazing on diazotrophs as a pathway for DDN into the zooplankton food
web. A caveat of our sampling for the qPCR study was a prolonged period (~ 6 h) between sample
collection and -80$^{\circ}$C freezing. Although the samples were stored damp and in an ice container
prior to freezing, it is likely that at least some gut evacuation would have occurred because the
samples were not anesthetized immediately upon collection (Gannon and Gannon, 1975).
Moreover, the qPCR assays were highly specific for their respective targets and as such, if the
animals consumed other targets (i.e. other diazotrophs or non-diazotrophs) these would not have
been detected or quantified. Finally, DNA extraction is not 100 % and underestimation of the
targets was therefore also possible.
However, the results from the qPCR assays do provide qualitative insights into zooplankton
ingestion of the targeted diazotrophs, and prey selection. All four of the qPCR targeted diazotrophs
(*Trichodesmium* spp., het-1, het-2, UCYN-C) were found in zooplankton guts. Overall, the most
frequently detected targets were het-1 and UCYN-C. Het-1 was most frequently detected in the
zooplankton during P1 and the beginning of P2, when this group dominated the diazotroph
community (Turk-Kubo et al., 2015). Similarly, UCYN-C was most frequently detected in the
zooplankton during P2, consistent with the UCYN-C bloom observed during that period. Although
target occurrence in the zooplankton largely reflected the prevalence of the diazotroph in the water
column, high detection was also recorded outside of periods of peak diazotroph occurrence. For
example, the highest abundance (277 *nifH* copies / copepod) for the *Trichodesmium* spp. target



measured by qPCR was on day 16 in M2, despite low water column abundance of this diazotroph
at that time; and het-2 was typically bd with the exception of day 2 when 277 *nifH* copies / copepod
were measured, again despite having low water column abundance at that time. This indicates that
the generally low abundance of *Trichodesmium* spp. and het-2 may have been due in part to top
down control through zooplankton grazing.
The $^{15}N_2$ labeled grazing experiments supported direct zooplankton grazing on UCYN-C, and
assimilation of ingested UCYN-C-derived N. Conversely, weak if any assimilation of DDN was
measured in the experiments where the diazotroph community was dominated by het-1 and
*Trichodesmium* spp.. This was a surprising finding given that het-1, and to a lesser extent
*Trichodesmium* spp., was detected in high abundance in copepod guts. A contributing factor to the
apparent low direct het-1 and *Trichodesmium* spp. DDN uptake may have been a lower atomic
enrichment of these diazotrophs. Indeed, the atomic enrichment of UCYN-C was more than double
that of *Trichodesmium* spp. in this experiment. Unfortunately the atomic enrichment of het-1 was
not measured and thus could not be assessed as a factor in the low to zero atomic enrichment of
the copepods in E1. Another contributing factor may have been variable encounter rates of
zooplankton with diazotroph prey. The total diazotroph abundance levels at the start of E2 and E4
were double (~ $3.6 \times 10^5$ and $4.5 \times 10^5$ *nifH* copies $L^{-1}$ respectively) those of E1 ($1.5 \times 10^5$ *nifH* copies
$L^{-1}$). Lower zooplankton encounter rates with het-1 may therefore have been a factor in the low
rate of DDN uptake during E1. Overall, therefore, questions remain as to the efficiency of direct
assimilation of het-1 and *Trichodesmium* spp. DDN by zooplankton. However, low to zero atomic
enrichment of zooplankton in E1, despite a 72 hour incubation, and previous observations that the
filamentous *Trichodesmium* spp. may not be easily digested by zooplankton (O'Neil and Roman,
1992), do suggest that indirect pathways of *Trichodesmium* spp. and het-1 DDN (through, e.g.,
microzooplankton or non-diazotrophic phytoplankton utilizing the dissolved DDN pool) to the
zooplankton are likely to be important.
As far as we are aware, this study provides the first evidence of direct zooplankton grazing on
UCYN-C. The average size of UCYN-C cells during VAHINE (5.7 µm) was on the lower end of
the spectrum effectively grazed by copepods, the dominant zooplankton during the experiment
(Fortier et al., 1994). However, an observation during the VAHINE experiment was that the
majority of the UCYN-C existed as aggregates (100-500 µm in size), likely making them more
accessible to these grazers (Bonnet et al., in review). During VAHINE it was estimated that ~ 16



% of total fixed $N_2$ during the UCYN-C bloom period was released to the dissolved pool, of which
~ 20 % was transferred to non-diazotrophic phytoplankton within 24 h (Bonnet et al., in review).
Therefore, although direct grazing on UCYN-C was demonstrated in this study, it is likely that
secondary pathways were also important in UCYN-C DDN transfer to zooplankton. Notably, the
largest decline in zooplankton $\delta^{15}N$ during VAHINE was observed during the UCYN-C bloom in
M1, further supporting an important contribution of UCYN-C-derived N to zooplankton biomass
in the New Caledonian lagoon.
**5    Conclusions**
The natural N isotope abundance of the zooplankton sampled during the VAHINE experiment
gave clear evidence for the importance of DDN to the zooplankton food web in the oligotrophic
south west New Caledonian lagoon. The mean DDN contribution to zooplankton biomass at the
start of the experiment was ~ 30 % indicating that the natural summer peak in diazotroph
production in this region was already contributing significantly to the lagoon plankton food web.
Stimulation of $N_2$ fixation rates in the VAHINE mesocosms corresponded with a weak
enhancement of DDN contribution to zooplankton biomass. This DDN contribution peaked at ~
70 % in M1 which is on the high end of estimates from other regions.
qPCR analysis, targeting four of the common diazotroph groups present during VAHINE
(*Trichodesmium* spp., het-1, het-2, UCYN-C), demonstrated that all were ingested by copepod
grazers. The most frequently detected targets were het-1 and UCYN-C, and their abundance in the
zooplankton corresponded with their periods of peak abundance in the mesocosms (P1 and P2
respectively). $^{15}N_2$ labeled grazing experiments provided evidence for direct ingestion and
assimilation of UCYN-C-derived N by the zooplankton, but not for het-1 and *Trichodesmium* spp..
We suggest that secondary pathways of *Trichodesmium* spp. and het-1 DDN to the zooplankton
are likely to be important.
As far as we are aware, this is the first reported instance of direct UCYN-C grazing by zooplankton.
Aggregation may make this small diazotroph more accessible to zooplankton grazers, however, in
the absence of aggregation, a high contribution to the dissolved pool, makes UCYN-C-derivedN
accessible to the zooplankton via secondary pathways. Through a combination of these N transfer
pathways it is evident that UCYN-C-derived N contributes significantly to the zooplankton food
web in the New Caledonia lagoon.




**Acknowledgements**
Funding for this research was provided by the Agence Nationale de la Recherche (ANR starting
grant VAHINE ANR-13-JS06-0002), the INSU-LEFE-CYBER program, GOPS and IRD. The
authors thank the captain and crew of the R/V *Alis*. We acknowledge the SEOH diver service from
IRD Noumea as well as the technical service of the IRD research center of Noumea for their helpful
technical support together with C. Guieu, J.-M. Grisoni and F. Louis from the Observatoire
Oceanologique de Villefranche-sur-mer (OOV) for the mesocosm design and the useful advice.
During this project B. Hunt was funded from the European Union's Seventh Framework
Programme for research, technological development and demonstration under grant agreement no.
302010 – project ISOZOO. The qPCR work was sponsored by a Knut and Alice Wallenberg
Foundation grant to R. Foster. B. Conroy was supported by NSF project DGE-0840804 and NSF
project OCE-0934036 awarded to D. Steinberg.

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





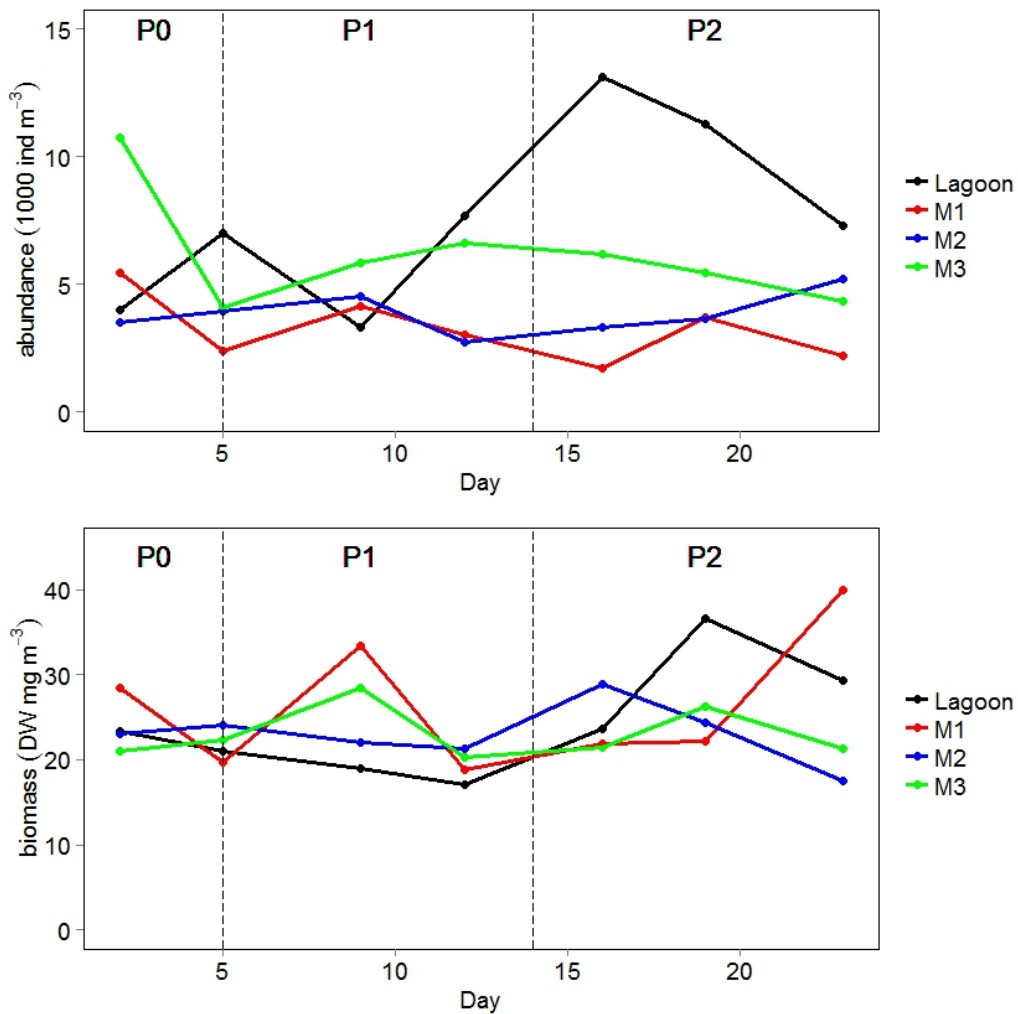

Figure 1. Zooplankton abundance (ind m$^{-3}$; above) and biomass (mg DW m$^{-3}$; below) over the 23
day VAHINE experiment (13 January to 4 February 2013) for the three VAHINE mescosms
(M1-3) and the lagoon waters. P0, P1 and P2 refer to the pre-phosphorous fertilization, DDA
dominated and UCYN-C dominated periods of the experiment respectively.







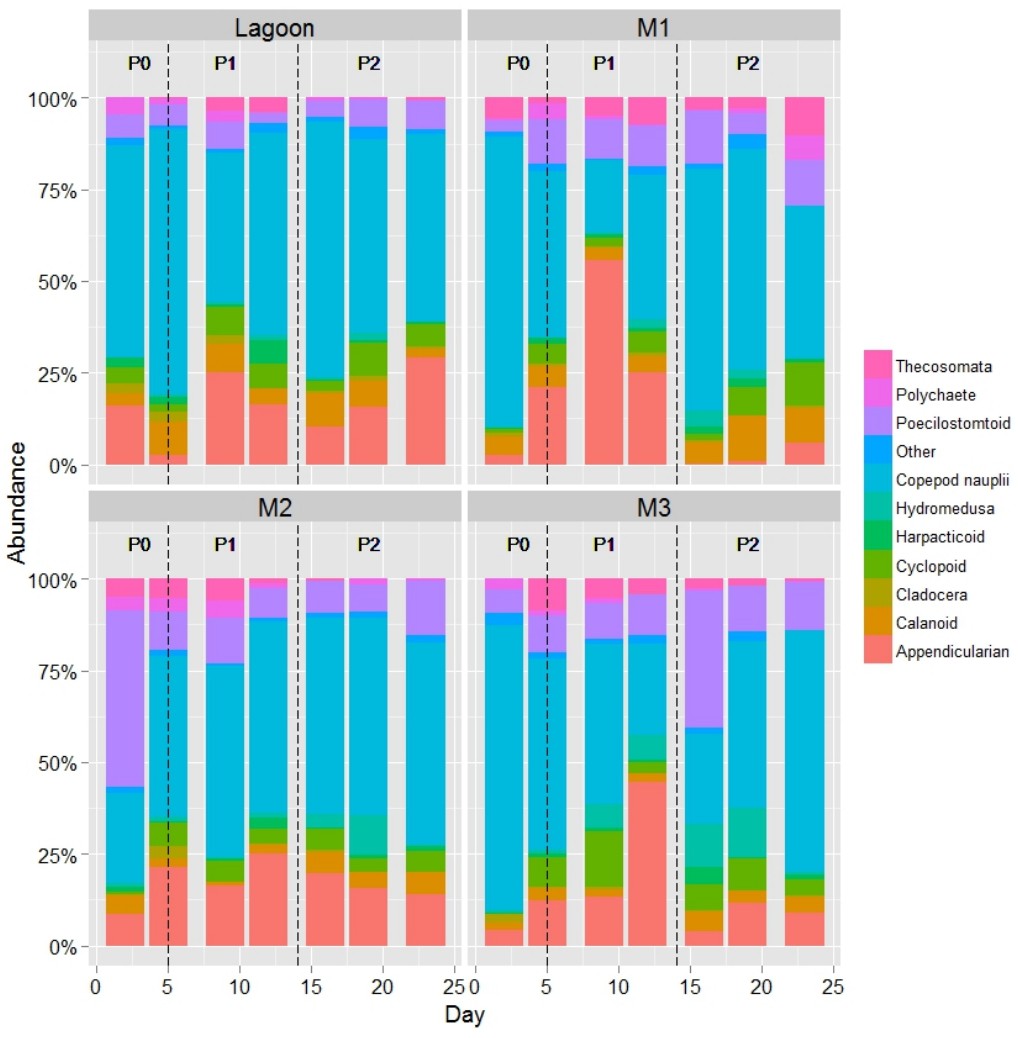

3    Figure 2. Proportional composition of zooplankton groups to total zooplankton abundance in the

4    three VAHINE mescosms (M1-3) and the lagoon waters.



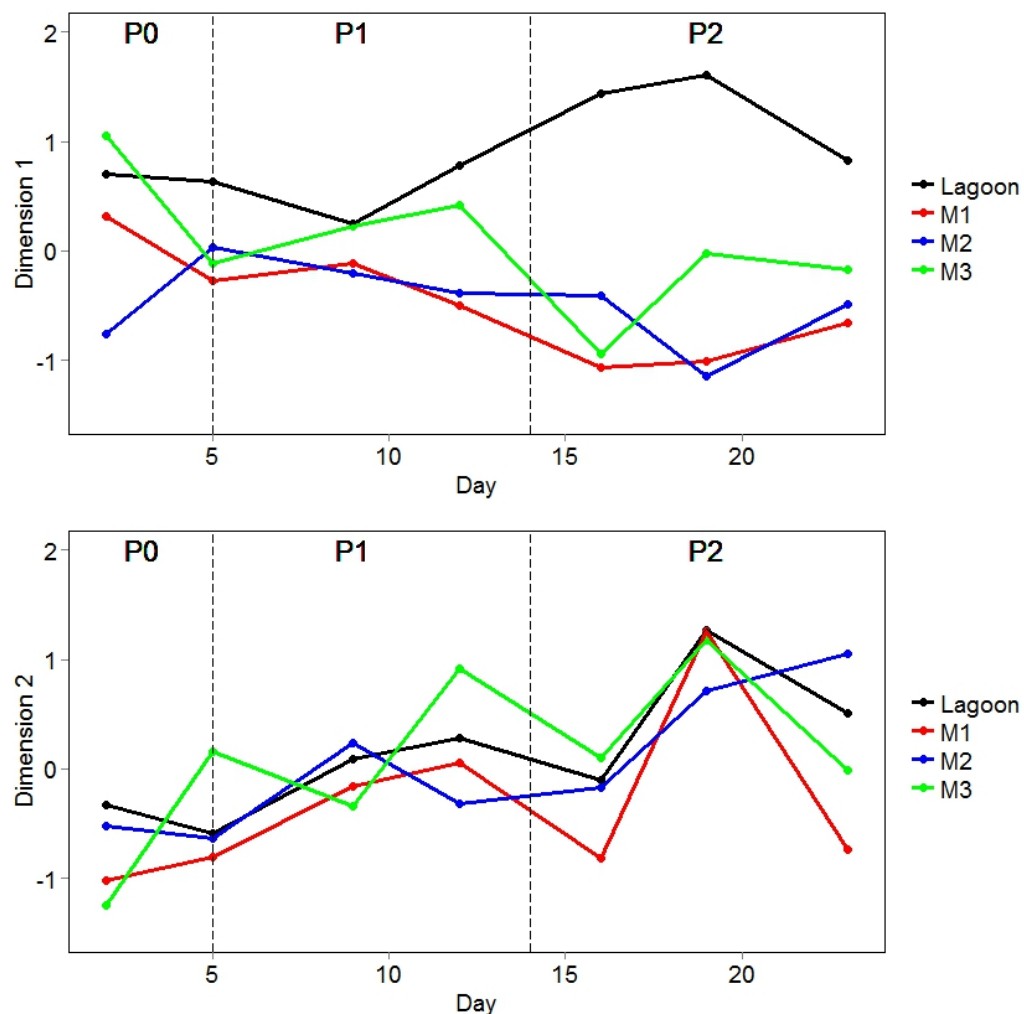

Figure 3. Zooplankton community NMDS ordination scores (Dimension 1 above and Dimension
2 below), based on Bray-Curtis similarity of fourth root transformed abundance data, over the 23
day VAHINE experiment (13 January to 4 February 2013) for the three VAHINE mescosms
(M1-3) and the lagoon waters. P0, P1 and P2 refer to the pre-phosphorous fertilization, DDA
dominated and UCYN-C dominated periods of the experiment respectively.





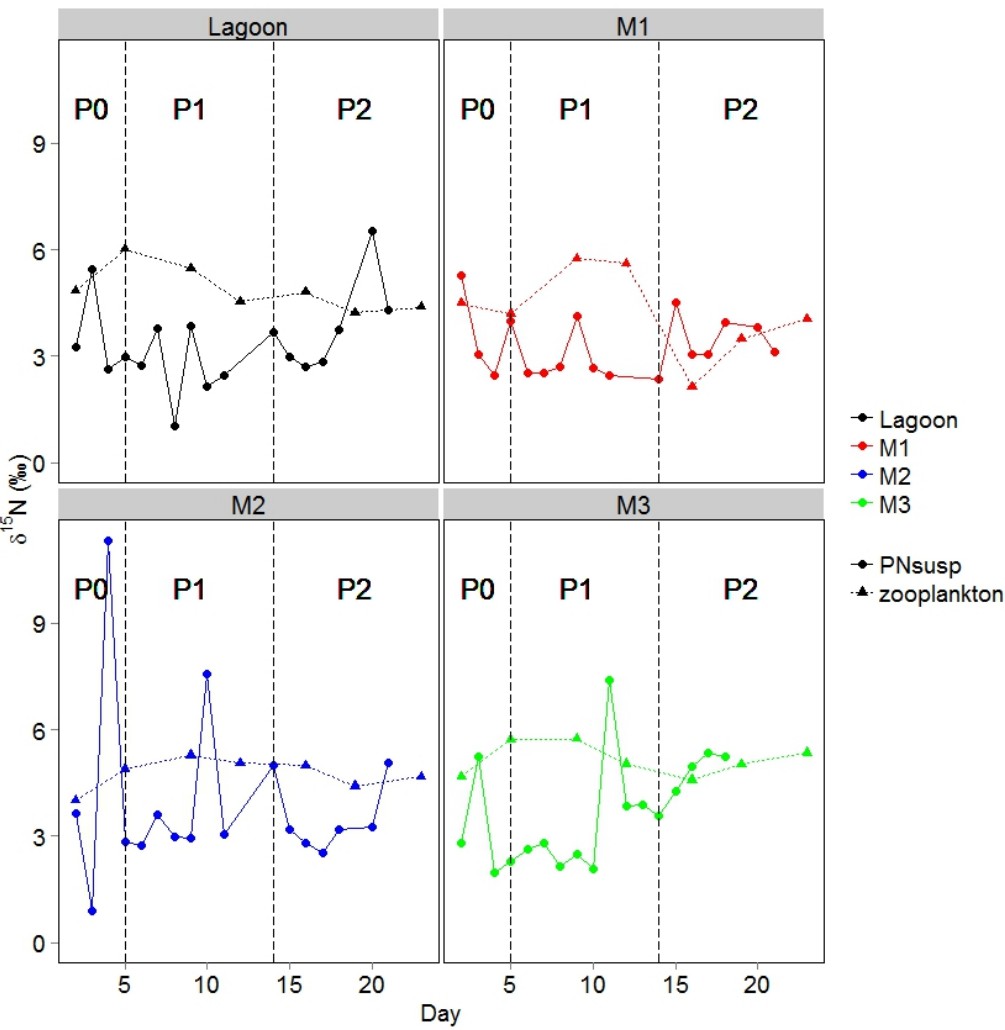

Figure 4. Nitrogen isotope ($\delta^{15}N$) values of zooplankton and suspended Particulate Nitrogen
($PN_{susp}$) over the course of the 23 day VAHINE experiment (13 January to 4 February 2013) for
the three VAHINE mesocosms (M1-3) and the lagoon waters. P0, P1 and P2 refer to the pre-
phosphorous fertilization, DDA dominated and UCYN-C dominated periods of the experiment
respectively. Zooplankton values are indicated by a solid lane and $PN_{susp}$ by a dashed line.



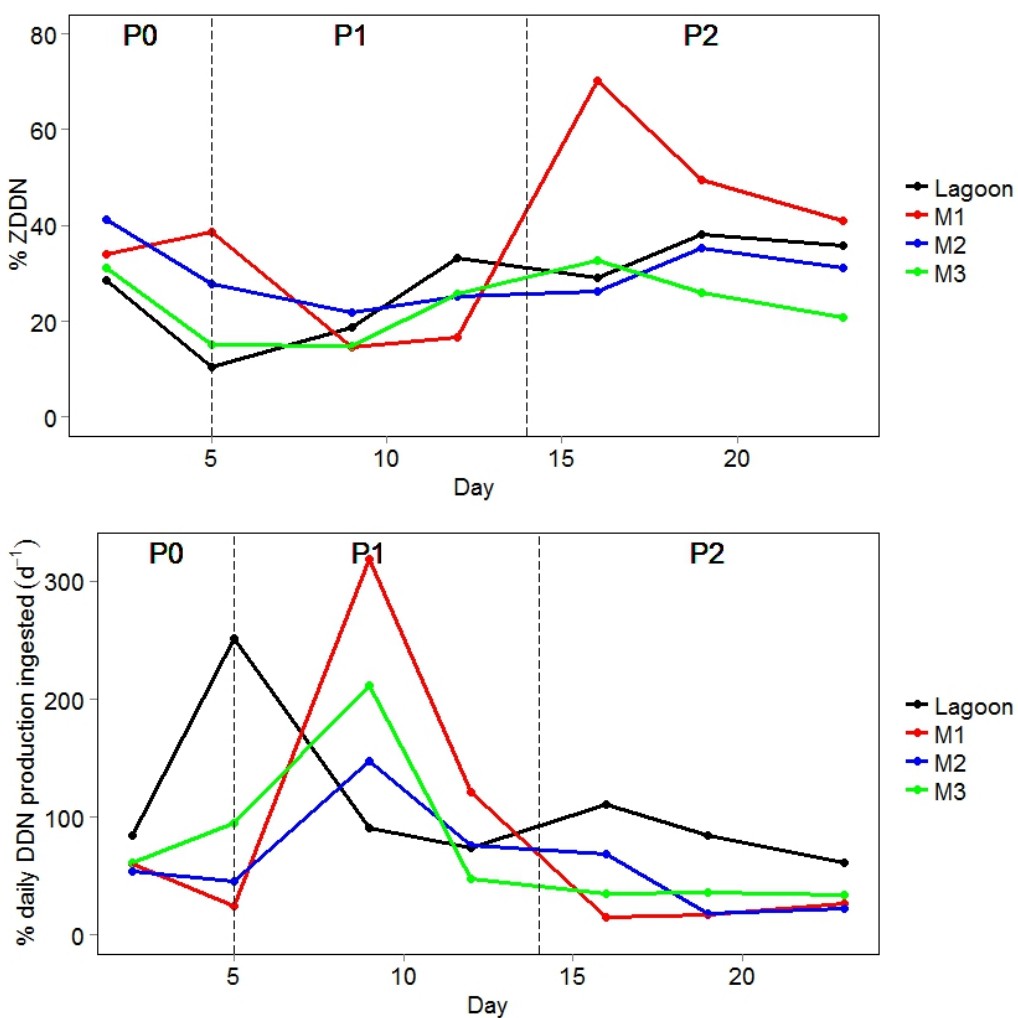

Figure 5. Percent contribution of diazotroph derived nitrogen (DDN) to zooplankton biomass

(above) and percent fixed nitrogen ingested by zooplankton.day[-1] over the course of the 23 day

VAHINE experiment (13 January to 4 February 2013) for the three mesocosms (M1-3) and the

lagoon waters. P0, P1 and P2 refer to the pre-phosphorous fertilization, DDA dominated and

UCYN-C dominated periods of the experiment respectively.



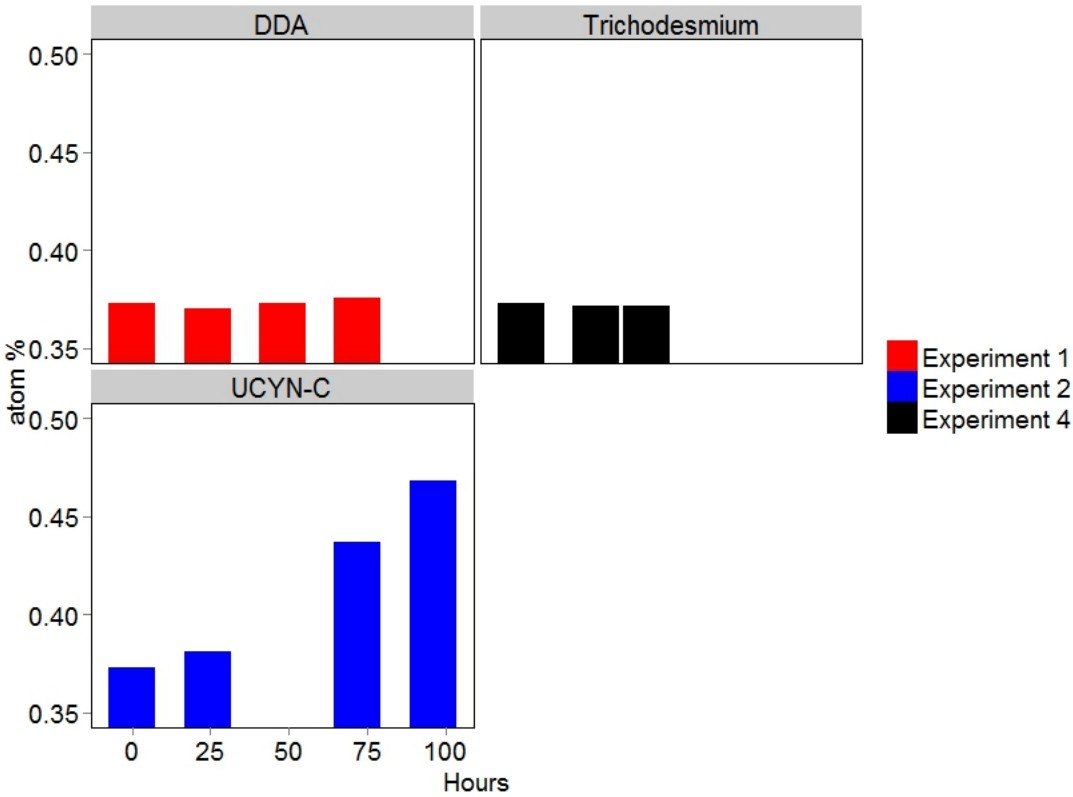



Figure 6. Atomic % enrichment of zooplankton in three $^{15}N_2$ labeled diazotroph grazing
experiments. The dominant diazotrophs in Experiments 1, 2 and 4 were DDA (het-1: *Richelia*
associated with *Rhizosolenia*), UCYN-C, and *Trichodesmium* spp. respectively. Zooplankton T0
atomic % enrichment was measured in triplicate for E1 and the average value was used as the
baseline for E1, E2 and E4. The atomic enrichment of the diazotroph community after 24 h was
1.515 % for UCYN-C and 0.613 % for *Trichodesmium* spp.. No enrichment value was obtained
for DDA.





Table 1. Summary of copepod samples processed for qPCR, targeting *Trichodesmium* spp., het-1 and the het-2 (DDA), and unicellular
group C (UCYN-C). All copepods per sample were pooled during the DNA extraction protocol. Site refers to the three VAHINE
mesocosms (M1-3) and the lagoon waters (La). The shading separates experimental periods P0, P1, and P2, corresponding with the
pre-phosphorous fertilization, DDA (het-1) dominated and UCYN-C dominated periods of the experiment respectively. het-1 =
*Richelia* associated with *Rhizosolenia*; het-2 = *Richelia* associated with *Hemiaulus*; bd = below detection; dnq = detectable but not
quantifiable; number in parenthesis = number of targets hit in 3 replicates.
*Table 1 overleaf...*



Table 1.

| Sample ID | Day | Site | Total no. copepods (n) | Calanoid (n) | Cyclopoid (n) | Harpacticoid (n) | het-1 *nifH* copies/ copepod | het-2 *nifH* copies/ copepod | *Trichodesmium* *nifH* copies/ copepod | UCYN-C *nifH* copies/ copepod |
|---|---|---|---|---|---|---|---|---|---|---|
| V3 | 2 | M2 | 35 | 13 | 13 | 9 | 173.31 | 62.14 | 264.4 | dnq (1) |
| V4 | 2 | La | 22 | 10 | 7 | 5 | bd | dnq (1) | bd | bd |
| V10 | 5 | M3 | 21 | 11 | 7 | 3 | bd | bd | bd | bd |
| V11 | 5 | M2 | 7 | 2 | 3 | 2 | bd | bd | bd | bd |
| V17 | 9 | M3 | 20 | 12 | 6 | 2 | dnq (1) | bd | bd | dnq (1) |
| V18 | 9 | M2 | 31 | 10 | 16 | 5 | dnq (1) | bd | bd | dnq (2) |
| V19 | 9 | M1 | 20 | 9 | 6 | 5 | 47.17 | bd | bd | 49.87 |
| V20 | 9 | La | 26 | 10 | 13 | 3 | 16.52 | bd | bd | bd |
| V25 | 12 | M3 | 22 | 7 | 10 | 5 | dnq (1) | bd | bd | bd |
| V26 | 12 | M2 | 29 | 11 | 9 | 9 | 34.83 | dnq (1) | bd | dnq (1) |
| V34 | 16 | M2 | 18 | 5 | 8 | 6 | 181.37 | n/a | 277.94 | 6.48 |
| V35 | 16 | M1 | 21 | 10 | 9 | 2 | bd | bd | dnq (1) | dnq (2) |
| V36 | 16 | La | 31 | 16 | 12 | 3 | 128.92 | bd | dnq (1) | dnq (1) |
| V41 | 19 | M3 | 27 | 15 | 9 | 3 | 26.84 | bd | dnq (1) | dnq(2) |
| V44 | 19 | La | 42 | 35 | 6 | 1 | dnq (2) | bd | bd | dnq(1) |
| V49 | 23 | M3 | 15 | 9 | 5 | 1 | dnq (1) | bd | dnq (2) | bd |
| V50 | 23 | M2 | 12 | 7 | 3 | 2 | bd | bd | bd | dnq (2) |
| V51 | 23 | M1 | 11 | 6 | 4 | 1 | bd | bd | bd | 28.72 |
| V52 | 23 | La | 20 | 9 | 4 | 7 | dnq (1) | bd | dnq (2) | 4.58 |




Table 2. Summary of three $^{15}N_2$ labeled diazotroph grazing experiments.

| | | | Number of zooplankton analysed | | | | |
|---|---|---|---|---|---|---|---|
| **Experiment** | **Day** | **Dominant diazotroph** | **0H** | **24H** | **48H (40H)** | **72H** | **96H** |
| E1 | 12 | DDA | 70 | 45 | 36 | 15 | |
| E2 | 17 | UCYN-C | | 90 | | 57 | 28 |
| E4 | 23 | *Trichodesmium* spp. | | 37 | (15) | | |









