# Peer review of "Contribution and pathways of diazotroph derived nitrogen to"

_Biogeosciences, 2015_

## Referee Comment (RC1) · Anonymous Referee #1 · 18 Feb 2016

Review of the manuscript Biogeosciences Discuss., bg-2015-614

Contribution and pathways of diazotroph derived nitrogen to zooplankton during the VAHINE mesocosm experiment in the oligotrophic New Caledonia lagoon

Authors: Brian P. V. Hunt Sophie Bonnet, Hugo Berthelot, Brandon J. Conroy, Rachel A. Foster, Marc Pagano

The manuscript by Hunt et al. is part of a series of manuscripts linked to the VAHINE mesocosm, dealing in this case with the transfer of nitrogen derived from N2 fixation to zooplankton over a 23 day period. I greatly appreciate the effort of this large scale

mesocosm experiment and its scientific objectives. I also greatly welcome the integration of gut measurements to identify diazotrophs ingested.

This manuscript is in general very well prepared and written. Moreover, experimental procedure and concept are thoroughly planned.

I only have some minor specific suggestions:

Abstract and Introduction

As I understood, the scope of the manuscript and experiment is to provide a time series and temporal variability in N2 fixation rates. This should be mentioned already in the abstract.

What does the abbreviation VAHINE stand for? Please add!

1. Page 3, line 24: Strange wording, please re-write e.g. the identification of the predominating pathway still in question.

2. Please add a list of accompanied manuscripts which deal with the VAHNE mesocosm experiment and their individual scope (I understand that there were a couple more).

Material and Methods

3. Page 5. I would restructure the first paragraph and make separate subheadings for Mesocosm description and Zooplankton sampling and processing

4. Page 6, line 24. Add counting error of enumeration.

5. Page 8, lines 23 ff. I doubt that the authors really determined direct grazing using the 15N set-up as it is presented. The microbial loop was likely still present in the incubation and recycling via bacteria attached to substrates and bacteriovorous nano- and microzooplankton might have occurred. Also see comment 13. Direct grazing nevertheless was truly identified via gut content analysis.

[Figure]

6. Was zooplankton put in non-labeled food after incubations so that they could purge their guts of non-digested N2 food? If not the measured N might overestimate nitrogen incorporation.

7. How many zooplankton species were pooled for the mass spectrometer analysis?

8. Also please provide a scheme for experiments and incubation that had been carried out.

9. Page 10, line 19. Why did you use a theoretical value for diazotrophs of -2‰ not the one measured during the VAHNE experiment?

Results and Discussion

10. Page 11, line 13 ff. It may be helpful to add a supplemental graph with phytoplankton data.

11. Page 15, line 4. Please change grazing to e.g. incorporation, as you did not determine direct grazing using the 15N tracer. See also comment number 5 (the authors also stated on page 19, line 4 "that secondary pathways were also important".

12. Figure 3. Why not show the actual nMDS plot, instead of showing nMDS dimensions versus time.

13. Figure 6. Please add label and numbers to the x- axis for Trichodesmium.
* * *

---

## Referee Comment (RC2) · Anonymous Referee #2 · 22 Feb 2016

General comments

Several studies have indicated DDN can significantly contribute to the food web base of zooplankton in systems where diazotrophs are important. Using a stable isotope approach, Montoya et al. (2002) found that the contribution of DDN to the food web base in the oligotrophic North Atlantic Ocean ranged from 0 – 67%. Rolff (2000) also found utilization of fixed N (DDN) by the zooplankton community in summer in the Baltic Sea. However questions remain as to the exact mechanisms whereby DDN enters the zooplankton food web. Many studies consider indirect paths, that is, diazotroph release

of DIN and DON (Capone et al., 1994; e.g., Ploug et al., 2011) and uptake of this N by the microbial loop, to be the major mechanism of DDN contribution to zooplankton. Evidence of direct grazing on diazotrophs has been more elusive, and has been considered limited due to a number of factors including toxicity of cyanobacteria (Sellner, 1997).

The study by Hunt et al. represents an advance in that it demonstrates using qPCR that zooplankton ingest many diazotrophs (at least the Trichodesmium spp., het-1, het-2, and UCYN-C present in their experiments). They also demonstrate for the first time using 15N labeling experiments the direct ingestion and assimilation of DDN from UCYN-C, but little assimilation of DDN from Trichodesmium spp. or het-1. Unicellular cyanobacteria (e.g., UCYN-C) can have abundances and N2 fixation rates greater than the more traditionally considered Trichodesmium spp. (Moisander et al., 2010), but few studies have examined the potential transfer of this new nitrogen to zooplankton. Thus this study indicates grazing of UCYN-C by zooplankton may be an important mechanism for transfer of DDN up the marine food web.

Hunt et al. also quantify the contribution of DDN to the base of the zooplankton food web using a two-endmember mixing model based on zooplankton $\delta$15N values throughout the mesocosm experiment. This is a powerful approach, and has been used successfully in several studies, however there are a few issues.

First, errors should be considered in the mixing model. The model makes several assumptions concerning endmembers (page 10 lines 17-22). Namely, TEF is assumed to be 2.2‰ the N isotope composition of diazotrophs is assumed to be $-2$‰ and a $\delta$15N value for zooplankton assuming a solely nitrate-based food web assumed to be 4.5‰ (nitrate) + 2.2‰ (TEF) = 6.7‰What are the errors on these estimates and how do they propagate into the final %DDN contribution? Diazotroph $\delta$15N values range between -1 to -2‰ for example (Montoya et al., 2002). The TEF of consumers raised on plant and algal diets is 2.2 $\pm$ 0.3‰ (McCutchan Jr. et al., 2003). However no errors are reported for %ZDDN (Figure 5), and thus the significance of the increase %ZDDN

over the experiment (page 16 lines 30-31) is not clear. Similarly, what are the errors associated with the calculation of % daily DDN production ingested (Figure 5)?

A more difficult issue is in the choice of the reference endmember for the mixing model. The reference endmember is the $\delta15N$ value for zooplankton assuming a solely nitrate-based food web, here assumed to be 4.5‰ (the $\delta15N$ value of nitrate entering the system) + 2.2‰ (TEF) = 6.7‰ for reference zooplankton. However the study site in New Caledonia is a LNLC system where recycled nutrients, e.g., NH4+, are likely important for production. Thus the actual reference endmember should be zooplankton $\delta15N$ values assuming recycling of new NO3- entering the system. This recycling will result in 15N depleted NH4+ and consequently zooplankton $\delta15N$ values that are lower than the assumed $\delta15N$-NO3- + TEF = 6.7‰.E.g., reference zooplankton $\delta15N$ values in Montoya et al. (2002) ranged from 4.3 – 6.4‰.The authors need to address how their choice of reference endmember affect %ZDDN, given recycling within the system.

Specific comments

1. P.2 line 15 – I find the phrase "% contribution of DDN to zooplankton biomass" somewhat confusing as it sounds like DDN is increasing zooplankton biomass. However this has been used in several studies (Montoya et al., 2002). The authors may want to consider if there is another phrase that may be more appropriate. 2. P.2 line 17 – What is BNF? 3. P.2 lines 21-24 – Consider rewriting this to make it more clear that all diazotrophs were ingested but only UCYN-C was assimilated significantly by zooplankton. 4. P.3 line 7 – What is sustaining 50% of primary productivity? I think they mean N2 fixation, but it sounds like they mean upwelled NO3-. 5. P.3 line 14 – Here and throughout the manuscript "$\delta15N$" should be "$\delta15N$ value". 6. P.3 line 17 – This would be true only in systems where N2 fixation is important. Clarify this. Which systems? 7. P.4 line 19 – Reference for "reduced feeding and egg production. . . . . .when fed a mixed cyanobacteria diet"? 8. P.6 line 25-26 – Which poecilostomatoid copepods do you refer to? Do you mean all cyclopoids? E.g., http://copepodes.obs-banyuls.fr/en/? 9. P.7 line 11 – Report all $\delta15N$ values at the same sig fig throughout the study, e.g.,

0.1‰ and 0.2‰ i̇0. P.10 line 18 – Report TEF as 2.2‰ i̇1. P.10 line 19 – Sig fig of -2‰ 12. P.11 line 29 – P.12 line 1 – What do you mean? N2 fixation in lagoon lower than mesocosm? Clarify. 13. P.12 line 2 – What did not differ? 14. P.13 line 7 - Do you mean cyclopoid? 15. P.13 line 24 – Sig figs. 16. P.16 line 26 – Do you mean $\delta$15N values of zooplankton?

References Capone, D.G., Ferrier, M.D., Carpenter, E.J., 1994. Amino acid cycling in colonies of the planktonic marine cyanobacterium Trichodesmium thiebautii. Appl. Environ. Microbiol. 60, 3989–3995. McCutchan Jr., J.H., Lewis Jr., W.M., Kendall, C., McGrath, C.C., 2003. Variation in trophic shift for stable isotope ratios of carbon, nitrogen, and sulfur. Oikos 102, 378–390. Moisander, P.H., Beinart, R.A., Hewson, I., White, A.E., Johnson, K.S., Carlson, C.A., Montoya, J.P., Zehr, J.P., 2010. Unicellular Cyanobacterial Distributions Broaden the Oceanic N-2 Fixation Domain. Science 327, 1512–1514. doi:10.1126/science.1185468 Montoya, J.P., Carpenter, E.J., Capone, D.G., 2002. Nitrogen fixation and nitrogen isotope abundances in zooplankton of the oligotrophic North Atlantic. Limnol. Oceanogr. 47, 1617–1628. Ploug, H., Adam, B., Musat, N., Kalvelage, T., Lavik, G., Wolf-Gladrow, D., Kuypers, M.M.M., 2011. Carbon, nitrogen and OâĆĆ fluxes associated with the cyanobacterium Nodularia spumigena in the Baltic Sea. Isme J. 5, 1549–1558. doi:10.1038/ismej.2011.20 Rolff, C., 2000. Seasonal variation in d13C and d15N of size-fractionated plankton at a coastal station in the northern Baltic proper. Mar. Ecol. Prog. Ser. 203, 47–65. Sellner, K.G., 1997. Physiology, ecology, and toxic properties of marine cyanobacteria blooms. Limnol. Oceanogr. 45, 1089–1104.

---

## Author Comment (AC1) · 19 Apr 2016

Anonymous Referee #1 This manuscript is in general very well prepared and written. Moreover, experimental procedure and concept are thoroughly planned.

I only have some minor specific suggestions:

Abstract and Introduction

As I understood, the scope of the manuscript and experiment is to provide a time series and temporal variability in N2 fixation rates. This should be mentioned already in the

abstract. What does the abbreviation VAHINE stand for? Please add!

Author: We will add the full program name: VAriability of vertical and tropHIc transfer of fixed N2 in the south wEst Pacific (VAHINE).

1. Page 3, line 24: Strange wording, please re-write e.g. the identification of the predominating pathway still in question.

Author: We will rewrite this as "...the predominant pathways of DDN into marine food webs are still in question (Wannicke et al., 2013)."

2. Please add a list of accompanied manuscripts which deal with the VAHNE mesocosm experiment and their individual scope (I understand that there were a couple more).

Author: We are not sure what the suggestion is here. That this list be added to the manuscript? This seems redundant given that our manuscript would be located in the special issue that houses all of the VAHINE manuscripts. We include citations of all of the relevant VAHINE manuscripts throughout our paper.

Material and Methods

3. Page 5. I would restructure the first paragraph and make separate subheadings for Mesocosm description and Zooplankton sampling and processing

Author: We will make separate subheadings for Mesocosm description and Zooplankton sampling and processing.

4. Page 6, line 24. Add counting error of enumeration.

Author: Following (Gifford and Caron, 2000) we estimated an enumeration error of 6.4% which we will add to the methods.

5. Page 8, lines 23 ff. I doubt that the authors really determined direct grazing using the 15N set-up as it is presented. The microbial loop was likely still present in the

incubation and recycling via bacteria attached to substrates and bacteriovorous nano and microzooplankton might have occurred. Also see comment 13. Direct grazing nevertheless was truly identified via gut content analysis.

Author: We agree that grazing in this case may have included ingestion of bacteria attached to substrates and bacteriovorous nano and microzooplankton. We will highlight these pathways in the text.

6. Was zooplankton put in non-labeled food after incubations so that they could purge their guts of non-digested N2 food? If not the measured N might overestimate nitrogen incorporation.

Author: No, zooplankton were not allowed to clear their guts. We will note this in the revised manuscript, as a source of potential overestimation of diazotroph nitrogen incorporation.

7. How many zooplankton species were pooled for the mass spectrometer analysis?

Author: As highlighted in the methods, we did not identify zooplankton to species level, but rather to order. The contribution of orders to samples is detailed on page 9, paragraph 2 and section 3.2.

8. Also please provide a scheme for experiments and incubation that had been carried out.

Author: We will provide a scheme of the experimental structure.

9. Page 10, line 19. Why did you use a theoretical value for diazotrophs of -2‰ not the one measured during the VAHNE experiment?

Author: We used a theoretical value for diazotrophs as we did not directly measure diazotrpoh $\delta$15N during VAHINE. Following the suggestion of Reviewer 2 we have amended this to include the diazotroph $\delta$15N range of -2 to -1 cited by Montoya et al. (2002), and used this range to estimate propagation of error.

[Figure]

Results and Discussion

10. Page 11, line 13 ff. It may be helpful to add a supplemental graph with phytoplankton data.

Author: Phytoplankton data are reported on extensively in Turk-Kubo et al., 2015. To avoid redundancy we briefly describe these authors findings and include reference to this paper.

11. Page 15, line 4. Please change grazing to e.g. incorporation, as you did not determine direct grazing using the 15N tracer. See also comment number 5 (the authors also stated on page 19, line 4 "that secondary pathways were also important".

Author: We will change grazing to uptake.

12. Figure 3. Why not show the actual nMDS plot, instead of showing nMDS dimensions versus time.

Author: We plotted the nMDS values against time to be able to more clearly illustrate how the zooplankton community developed with time. A unit-less nMDS plot would require labelling of all dates for all samples, presenting a more cluttered view of the time series.

13. Figure 6. Please add label and numbers to the x- axis for Trichodesmium.

Author: We will add label and numbers to the x- axis for Trichodesmium.

---

## Author Comment (AC2) · 19 Apr 2016

General comments Several studies have indicated DDN can significantly contribute to the food web base of zooplankton in systems where diazotrophs are important. Using a stable isotope approach, Montoya et al. (2002) found that the contribution of DDN to the food web base in the oligotrophic North Atlantic Ocean ranged from 0 – 67%. Rolff (2000) also found utilization of fixed N (DDN) by the zooplankton community in summer in the Baltic Sea. However questions remain as to the exact mechanisms

whereby DDN enters the zooplankton food web. Many studies consider indirect paths, that is, diazotroph release of DIN and DON (Capone et al., 1994; e.g., Ploug et al., 2011) and uptake of this N by the microbial loop, to be the major mechanism of DDN contribution to zooplankton. Evidence of direct grazing on diazotrophs has been more elusive, and has been considered limited due to a number of factors including toxicity of cyanobacteria (Sellner, 1997). The study by Hunt et al. represents an advance in that it demonstrates using qPCR that zooplankton ingest many diazotrophs (at least the Trichodesmium spp., het-1, het- 2, and UCYN-C present in their experiments). They also demonstrate for the first time using 15N labeling experiments the direct ingestion and assimilation of DDN from UCYN-C, but little assimilation of DDN from Trichodesmium spp. or het-1. Unicellular cyanobacteria (e.g., UCYN-C) can have abundances and N2 fixation rates greater than the more traditionally considered Trichodesmium spp. (Moisander et al., 2010), but few studies have examined the potential transfer of this new nitrogen to zooplankton. Thus this study indicates grazing of UCYN-C by zooplankton may be an important mechanism for transfer of DDN up the marine food web.

Hunt et al. also quantify the contribution of DDN to the base of the zooplankton food web using a two-endmember mixing model based on zooplankton _15N values throughout the mesocosm experiment. This is a powerful approach, and has been used successfully in several studies, however there are a few issues.

First, errors should be considered in the mixing model. The model makes several assumptions concerning endmembers (page 10 lines 17-22). Namely, TEF is assumed to be 2.2‰ the N isotope composition of diazotrophs is assumed to be -2‰ and a $\delta$15N value for zooplankton assuming a solely nitrate-based food web assumed to be 4.5‰ (nitrate) + 2.2‰ (TEF) = 6.7‰ What are the errors on these estimates and how do they propagate into the final %DDN contribution? Diazotroph $\delta$15N values range between -1 to -2‰ for example (Montoya et al., 2002). The TEF of consumers raised on plant and algal diets is 2.2 $\pm$ 0.3‰ (McCutchan Jr. et al., 2003). However no errors are reported for %ZDDN (Figure 5), and thus the significance of the increase %ZDDN

over the experiment (page 16 lines 30-31) is not clear. Similarly, what are the errors associated with the calculation of % daily DDN production ingested (Figure 5)?

Author: Calculation of error margins for our estimates of 1. diazotroph nitrogen contribution to zooplankton biomass and 2. % daily DDN production ingested consumed, is an important point. This will certainly improve the quality of our estimates. We have calculated the min and max values for both 1 and 2 taking into account the error in TEF estimate (2.2$\pm$ 0.3, following McClutchan et al 2003), and a range of diazotroph $\delta$15N values between -1 and -2‰ (following Montoya et al., 2002), using -1.5 as the mean value. The min and max values will be included as error bars for measures 1 and 2 in Figure 5.

A more difficult issue is in the choice of the reference endmember for the mixing model. The reference endmember is the $\delta$15N value for zooplankton assuming a solely nitrate based food web, here assumed to be 4.5‰ (the $\delta$15N value of nitrate entering the system) + 2.2‰ (TEF) = 6.7‰ for reference zooplankton. However the study site in New Caledonia is a LNLC system where recycled nutrients, e.g., NH4+, are likely important for production. Thus the actual reference endmember should be zooplankton $\delta$15N values assuming recycling of new NO3- entering the system. This recycling will result in 15N depleted NH4+ and consequently zooplankton $\delta$15N values that are lower than the assumed $\delta$15N -NO3- + TEF = 6.7‰ËŹ .g., reference zooplankton $\delta$15N values in Montoya et al. (2002) ranged from 4.3 – 6.4‰ŹThe authors need to address how their choice of reference endmember affect %ZDDN, given recycling within the system.

Author: We agree with Reviewer 2 that the choice of model end member is difficult issue, and that we did not detail this sufficiently in the first version of our manuscript. Perhaps the most challenging aspect of this is that although the New Caledonia lagoon is a LNLC environment it is also an environment apparently strongly influenced by nitrogen fixation. It is therefore not possible to confidently select zooplankton samples from the lagoon that will not reflect at least some influence of diazotrophic nitrogen. Indeed, Montoya et al (2002) noted this specifically as an issue in their study. Although

they used a $\delta$15N range of 4.3 – 6.4‰ as their zooplankton reference value, they noted:

"Because the reference zooplankton used in Eq. 2 may reflect some inputs of recently fixed nitrogen, the values shown in Table 2 are a conservative estimate of the role of diazotroph nitrogen in supporting zooplankton biomass production. In fact, measurements of the $\delta$15N values of individual amino acids isolated from zooplankton collected at selected stations of leg 2 of cruise SJ9603 are consistent with a higher diazotroph contribution, approaching 100% at times, to the zooplankton in the western part of the transect (McClelland et al. pers. comm.)." page 1625, paragraph 1.

In a previous paper (Hunt et al. 2015) we recorded mean zooplankton grazer $\delta$15N of 5.94‰ in the Low Nitrate Low Chlorophyll region east of New Caledonia. Given that there was likely some influence of diazotrophy in that region, the zooplankton end member value of 6.7‰ used in this study does seem to be a realistic estimate of $\delta$15N not influenced by diazotrophic nitrogen.

Specific comments

1. P.2 line 15 – I find the phrase "% contribution of DDN to zooplankton biomass" somewhat confusing as it sounds like DDN is increasing zooplankton biomass. However this has been used in several studies (Montoya et al., 2002). The authors may want to consider if there is another phrase that may be more appropriate.

Author: We will clarify this as "% contribution of DDN to zooplankton nitrogen biomass".

2. P.2 line 17 – What is BNF?

Author: This is a typo and has been removed from the abstract.

3. P.2 lines 21-24 – Consider rewriting this to make it more clear that all diazotrophs were ingested but only UCYN-C was assimilated significantly by zooplankton.

Author: We have re-worded these lines as "qPCR analysis targeting four of the common diazotroph groups present in the mesocosms (Trichodesmium, het-1, het-

2, UCYN-C) demonstrated that all four were ingested by copepod grazers, and that their abundance in copepod stomachs generally corresponded with their in situ abundance. 15N2 labeled grazing experiments therefore provided evidence for direct ingestion and assimilation of UCYN-C-derived N by the zooplankton, but not for het-1 and Trichodesmium, supporting an important role of secondary pathways of DDN to the zooplankton for the latter groups,..."

4. P.3 line 7 – What is sustaining 50% of primary productivity? I think they mean N2 fixation, but it sounds like they mean upwelled NO3-.

Author: We will clarify this sentence as follows:

"In the oligotrophic tropical and subtropical oceans, where strong stratification limits the upward mixing of nitrate replete deep water into the photic zone, this new N is particularly important, sustaining $\sim$50 % of primary productivity (Karl et al., 1997).

5. P.3 line 14 – Here and throughout the manuscript "$\delta$15N" should be "$\delta$15N value".

Author: We will make this change.

6. P.3 line 17 – This would be true only in systems where N2 fixation is important. Clarify this. Which systems?

Author: This refers to phytoplankton $\delta$15N where nitrate is the primary nitrogen source, i.e., "By comparison, the average ocean nitrate $\delta$15N is $\sim$ 5 ‰ (Sigman et al., 1999; Sigman et al., 1997), leading to higher $\delta$15N for primary producers using this source."

7. P.4 line 19 – Reference for "reduced feeding and egg production: : : : :when fed a mixed cyanobacteria diet"?

Author: Sellner et al (1996) Phycologia, 35, 177-182. We will add this reference.

8. P.6 line 25-26 – Which poecilostomatoid copepods do you refer to? Do you mean all cyclopoids? E.g., http://copepodes.obs-banyuls.fr/en/?

Author: Poecilostomatoid are a separate order, previously included with the Cyclopoids.

http://www.marinespecies.org/aphia.php?p=taxdetails&id=155879

9. P.7 line 11 – Report all _15N values at the same sig fig throughout the study, e.g., 0.1‰ and 0.2‰Ź

Author: Yes. This will be done.

10. P.10 line 18 – Report TEF as 2.2‰Ź

Author: Yes. This will be done.

11. P.10 line 19 – Sig fig of -2‰

Author: Yes. This will be done.

12. P.11 line 29 – P.12 line 1 – What do you mean? N2 fixation in lagoon lower than mesocosm? Clarify.

Author: This is clarified in the text as follows:

"N2 fixation rates measured in the lagoon waters were significantly (p<0.05) lower than those measured in lagoon waters (9.2±4.7 nmol N L-1 d-1) over the 23 days of the experiment."

13. P.12 line 2 – What did not differ?

Author: This will be clarified as follows:

"N2 fixation rates measured in the lagoon waters (average = 9.2±4.7 nmol N L-1 d-1) were significantly (p<0.05) lower than those measured in the mesocosm over the 23 days of the experiment."

14. P.13 line 7 - Do you mean cyclopoid?

Author: No, we mean poecilostomatoid.

15. P.13 line 24 – Sig figs. 16. P.16 line 26 – Do you mean $\delta$15N values of zooplankton?
Author: Correct, we will amend this to $\delta$15N values.

---

## Author Response (AR1)

*Reviewer comments in italics and bold*
Author response not in italics or bold.

**Anonymous Referee #1**

*Review of the manuscript Biogeosciences Discuss., bg-2015-614*

*Contribution and pathways of diazotroph derived nitrogen to zooplankton during the VAHINE mesocosm experiment in the oligotrophic New Caledonia lagoon*

*Authors: Brian P. V. Hunt Sophie Bonnet, Hugo Berthelot, Brandon J. Conroy, Rachel A. Foster, Marc Pagano*

*The manuscript by Hunt et al. is part of a series of manuscripts linked to the VAHINE mesocosm, dealing in this case with the transfer of nitrogen derived from N2 fixation to zooplankton over a 23 day period. I greatly appreciate the effort of this large scale mesocosm experiment and its scientific objectives. I also greatly welcome the integration of gut measurements to identify diazotrophs ingested.*

*This manuscript is in general very well prepared and written. Moreover, experimental procedure and concept are thoroughly planned.*

*I only have some minor specific suggestions:*

*Abstract and Introduction*

*As I understood, the scope of the manuscript and experiment is to provide a time series and temporal variability in N2 fixation rates. This should be mentioned already in the abstract. What does the abbreviation VAHINE stand for? Please add!*

We have added the full program name to the abstract and on first mentions in the Introduction: VAriability of vertical and tropHIc transfer of fixed N2 in the south wEst Pacific (VAHINE).

*1. Page 3, line 24: Strange wording, please re-write e.g. the identification of the predominating pathway still in question.*

We have re-written this as "…the predominant pathways of DDN into marine food webs are still in question (Wannicke et al., 2013)."

*2. Please add a list of accompanied manuscripts which deal with the VAHNE mesocosm experiment and their individual scope (I understand that there were a couple more).*

We are not sure what the suggestion is here. That this list be added to the manuscript? This seems redundant given that our manuscript would be located in the special issue that houses all

of the VAHINE manuscripts. We include citations of all of the relevant VAHINE manuscripts throughout our paper.

*Material and Methods*

*3. Page 5. I would restructure the first paragraph and make separate subheadings for Mesocosm description and Zooplankton sampling and processing*

We have made separate subheadings for Mesocosm description and Zooplankton sampling and processing.

*4. Page 6, line 24. Add counting error of enumeration.*

Following (Gifford and Caron, 2000) we estimated an enumeration error of 6.4%. We have added this to the methods – section 2.2., paragraph 3.

*5. Page 8, lines 23 ff. I doubt that the authors really determined direct grazing using the 15N set-up as it is presented. The microbial loop was likely still present in the incubation and recycling via bacteria attached to substrates and bacteriovorous nano and microzooplankton might have occurred. Also see comment 13. Direct grazing nevertheless was truly identified via gut content analysis.*

We agree that grazing in this case may have included ingestion of bacteria attached to substrates and bacteriovorous nano and microzooplankton. We have highlighted this in Section 2.4, paragraph 2:

"Since the role of the microbial loop in making diazotroph nitrogen available to the zooplankton was not determined, the experiments are indicative of diazotroph nitrogen uptake and incorporation by the zooplankton but not necessarily the pathways."

*6. Was zooplankton put in non-labeled food after incubations so that they could purge their guts of non-digested N2 food? If not the measured N might overestimate nitrogen incorporation.*

We have noted this in section 3,4, paragraph 1:

"It should be noted that zooplankton were not allowed to purge their stomach contents after the incubation experiments, and this may have been a source of overestimation of diazotroph nitrogen incorporation. However, the persistent increase during E2 does indicate that diazotroph nitrogen incorporation was the primary factor in observed atomic enrichment."

*7. How many zooplankton species were pooled for the mass spectrometer analysis?*

As highlighted in the methods, we did not identify zooplankton to species level, but rather to order. The contribution of orders to samples is detailed on page 9, paragraph 2 and section 3.2.

*8. Also please provide a scheme for experiments and incubation that had been carried out.*

We have provided a scheme of the experimental structure (see Figure 1).

*9. Page 10, line 19. Why did you use a theoretical value for diazotrophs of -2‰ not the one measured during the VAHNE experiment?*

We used a theoretical value for diazotrophs as we did not directly measure diazotroph $\delta^{15}N$ during VAHINE. Following the suggestion of Reviewer 2 we have amended this to include the diazotroph $\delta^{15}N$ range of -2 to -1 cited by Montoya et al. (2002), and used this range to estimate propagation of error.

*Results and Discussion*

*10. Page 11, line 13 ff. It may be helpful to add a supplemental graph with phytoplankton data.*

Phytoplankton data are reported on extensively in Turk-Kubo et al., 2015. To avoid redundancy we briefly describe these authors findings and inlcude reference to this paper.

*11. Page 15, line 4. Please change grazing to e.g. incorporation, as you did not determine direct grazing using the 15N tracer. See also comment number 5 (the authors also stated on page 19, line 4 "that secondary pathways were also important".*

We have changed the title of Section 3.4 to "Zooplankton incorporation of diazotroph nitrogen".

*12. Figure 3. Why not show the actual nMDS plot, instead of showing nMDS dimensions versus time.*

We plotted the nMDS values against time to be able to more clearly illustrate how the zooplankton community developed with time. A unit-less nMDS plot would require labelling of all dates for all samples, presenting a more cluttered view of the time series.

*13. Figure 6. Please add label and numbers to the x- axis for Trichodesmium.*

We have added label and numbers to the x- axis for each experiment.

**Anonymous Referee #2**

*General comments*
*Several studies have indicated DDN can significantly contribute to the food web base of zooplankton in systems where diazotrophs are important. Using a stable isotope approach, Montoya et al. (2002) found that the contribution of DDN to the food web base in the oligotrophic North Atlantic Ocean ranged from 0 – 67%. Rolff (2000) also found utilization of fixed N (DDN) by the zooplankton community in summer in the Baltic Sea. However questions remain as to the exact mechanisms whereby DDN enters the zooplankton food web.*

*Many studies consider indirect paths, that is, diazotroph release of DIN and DON (Capone et al., 1994; e.g., Ploug et al., 2011) and uptake of this N by the microbial loop, to be the major mechanism of DDN contribution to zooplankton. Evidence of direct grazing on diazotrophs has been more elusive, and has been considered limited due to a number of factors including toxicity of cyanobacteria (Sellner, 1997).*

*The study by Hunt et al. represents an advance in that it demonstrates using qPCR that zooplankton ingest many diazotrophs (at least the Trichodesmium spp., het-1, het-2, and UCYN-C present in their experiments). They also demonstrate for the first time using 15N labeling experiments the direct ingestion and assimilation of DDN from UCYN-C, but little assimilation of DDN from Trichodesmium spp. or het-1. Unicellular cyanobacteria (e.g., UCYN-C) can have abundances and N2 fixation rates greater than the more traditionally considered Trichodesmium spp. (Moisander et al., 2010), but few studies have examined the potential transfer of this new nitrogen to zooplankton. Thus this study indicates grazing of UCYN-C by zooplankton may be an important mechanism for transfer of DDN up the marine food web.*

*Hunt et al. also quantify the contribution of DDN to the base of the zooplankton food web using a two-endmember mixing model based on zooplankton _15N values throughout the mesocosm experiment. This is a powerful approach, and has been used successfully in several studies, however there are a few issues.*

*First, errors should be considered in the mixing model. The model makes several assumptions concerning endmembers (page 10 lines 17-22). Namely, TEF is assumed to be 2.2‰ the N isotope composition of diazotrophs is assumed to be -2‰ and a $\delta^{15}N$ value for zooplankton assuming a solely nitrate-based food web assumed to be 4.5‰ (nitrate) + 2.2‰ (TEF) = 6.7‰ What are the errors on these estimates and how do they propagate into the final %DDN contribution? Diazotroph $\delta^{15}N$ values range between -1 to -2‰ for example (Montoya et al., 2002). The TEF of consumers raised on plant and algal diets is 2.2 ± 0.3‰ (McCutchan Jr. et al., 2003). However no errors are reported for %ZDDN (Figure 5), and thus the significance of the increase %ZDDN over the experiment (page 16 lines 30-31) is not clear. Similarly, what are the errors associated with the calculation of % daily DDN production ingested (Figure 5)?*

Calculation of error margins for our estimates of 1. diazotroph nitrogen contribution to zooplankton biomass and 2. % daily DDN production ingested consumed, is an important point.

We have recalculated our %ZDDN as follows:

"TEF is the trophic enrichment factor, which was set at 2.2 ± 0.3 (McCutchan et al., 2003; Vanderklift and Ponsard, 2003); $\delta^{15}N_{diazo}$ is the isotopic signature of diazotrophs, for which we used a range of -1 to -2 ‰ (Montoya et al., 2002); $\delta^{15}N_{zplref}$ is the isotopic signature of zooplankton assuming nitrate based phytoplankton production, and for this we used a value of 6 ‰ from the ocean west of New Caledonia where nitrogen fixation is reduced (Hunt et al., 2015). Minimum, average and maximum % ZDDN were estimated using the lower, mean and upper bounds of TEF and the $\delta^{15}N_{diazo}$ values cited above." – Section 2.6, paragraph 1.

The minimum and maximum values have been added as error bars in Figure 5.

We also note that we have updated our P:B ratio used to estimate daily DDN ingestion, drawing from the overview of tropical plankton presented by (Le Borgne, 1987).

In re-calculating estimated daily DDN ingestion we encountered an error in our estimate due to the worksheet formula using the % ZDDN value as a numeric rather than a percentage. We have corrected this in the revised Figure 5. The paragraph detailing the calculation of daily DDN ingestion has been updated accordingly:

"where N content (mg DW) was calculated using a mean value of 4.25 % for a mixed zooplankton community in Uvea Lagoon (Le Borgne et al., 1997); daily zooplankton production (mg DW d$^{-1}$) was calculated using a Production: Biomass ratio of 37.5 % (Le Borgne, 1987); daily excretion was calculated assuming a net growth efficiency (K) of 0.513 (Le Borgne et al., 1997); and assimilation efficiency was set at 0.7 (Le Borgne et al., 1997). The range of daily DDN production ingested by zooplankton was estimated using the calculated minimum, average and maximum % ZDDN values"

We note that the extremely high estimated percent of daily DDN production ingested (averaging ~ 240 %) likely reflects the longer integration time of stable isotope measurements and accumulation of the DDN signature in the zooplankton over multiple days. Similar values were reported by (Montoya et al., 2002).

*A more difficult issue is in the choice of the reference endmember for the mixing model. The reference endmember is the $\delta^{15}N$ value for zooplankton assuming a solely nitrate based food web, here assumed to be 4.5‰ (the $\delta^{15}N$ value of nitrate entering the system) + 2.2‰ (TEF) = 6.7‰ for reference zooplankton. However the study site in New Caledonia is a LNLC system where recycled nutrients, e.g., NH4+, are likely important for production. Thus the actual reference endmember should be zooplankton $\delta^{15}N$ values assuming recycling of new NO3- entering the system. This recycling will result in 15N depleted NH4+ and consequently zooplankton $\delta^{15}N$ values that are lower than the assumed $\delta^{15}N$ -NO3- + TEF = 6.7‰E˙ .g., reference zooplankton $\delta^{15}N$ values in Montoya et al. (2002) ranged from 4.3 – 6.4‰˙The authors need to address how their choice of reference endmember affect %ZDDN, given recycling within the system.*

We agree with Reviewer 2 that the choice of model end member is difficult issue, and that we did not detail this sufficiently in the first version of our manuscript. Perhaps the most challenging aspect of this is that although the New Caledonia lagoon is a LNLC environment it is also an environment apparently strongly influenced by nitrogen fixation. It is therefore not possible to confidently select zooplankton samples from the lagoon that will not reflect at least some influence of diazotrophic nitrogen. Indeed, Montoya et al (2002) noted this specifically as an issue in their study. Although they used a $\delta^{15}N$ range of 4.3 – 6.4‰ as their zooplankton reference value, they noted:

"Because the reference zooplankton used in Eq. 2 may reflect some inputs of recently fixed nitrogen, the values shown in Table 2 are a conservative estimate of the role of diazotroph nitrogen in supporting zooplankton biomass production. In fact, measurements of the $\delta^{15}N$ values of individual amino acids isolated from zooplankton collected at selected stations of leg 2 of cruise SJ9603 are consistent with a higher diazotroph contribution, approaching 100% at times,

to the zooplankton in the western part of the transect (McClelland et al. pers. comm.)." page 1625, paragraph 1.

In a previous paper (Hunt et al. 2015) we recorded mean zooplankton grazer $\delta^{15}N$ of 5.94‰ in the Low Nitrate Low Chlorophyll region east of New Caledonia. Accordingly we have now changed the baseline value used in our study to 6 ‰ (see abaove).

***Specific comments***

***1. P.2 line 15 – I find the phrase "% contribution of DDN to zooplankton biomass" somewhat confusing as it sounds like DDN is increasing zooplankton biomass. However this has been used in several studies (Montoya et al., 2002). The authors may want to consider if there is another phrase that may be more appropriate.***

We have changed this to "% contribution of DDN to zooplankton nitrogen biomass".

***2. P.2 line 17 – What is BNF?***

This is a typo and has been removed from the abstract.

***3. P.2 lines 21-24 – Consider rewriting this to make it more clear that all diazotrophs were ingested but only UCYN-C was assimilated significantly by zooplankton.***

We have re-worded these lines as "qPCR analysis targeting four of the common diazotroph groups present in the mesocosms (*Trichodesmium*, het-1, het-2, UCYN-C) demonstrated that all four were ingested by copepod grazers, and that their abundance in copepod stomachs generally corresponded with their *in situ* abundance. $^{15}N_2$ labeled grazing experiments therefore provided evidence for direct ingestion and assimilation of UCYN-C-derived N by the zooplankton, but not for het-1 and *Trichodesmium*, supporting an important role of secondary pathways of DDN to the zooplankton for the latter groups,…"

***4. P.3 line 7 – What is sustaining 50% of primary productivity? I think they mean N2 fixation, but it sounds like they mean upwelled NO3-.***

We have clarified this sentence as follows:

"In the oligotrophic tropical and subtropical oceans, where strong stratification limits the upward mixing of nitrate replete deep water into the photic zone, this new N is particularly important, sustaining ~50 % of primary productivity (Karl et al., 1997).

***5. P.3 line 14 – Here and throughout the manuscript "$\delta^{15}N$" should be "$\delta^{15}N$ value".***

We have corrected this throughout the manuscript.

***6. P.3 line 17 – This would be true only in systems where N2 fixation is important. Clarify this. Which systems?***

This refers to phytoplankton $\delta^{15}N$ values where nitrate is the primary nitrogen source. We have clarified this as follows:

"By comparison, the average ocean nitrate $\delta^{15}N$ value is ~ 5 ‰ (Sigman et al., 1999; Sigman et al., 1997), leading to higher $\delta^{15}N$ values for primary producers using nitrate as their nitrogen source."

**7. P.4 line 19 – Reference for "reduced feeding and egg production when fed a mixed cyanobacteria diet"?**

Sellner et al (1996) Phycologia, 35, 177-182. We have added this reference.

**8. P.6 line 25-26 – Which poecilostomatoid copepods do you refer to? Do you mean all cyclopoids? E.g., http://copepodes.obs-banyuls.fr/en/?**

Poecilostomatoid are a separate order, previously included with the Cyclopoids.

http://www.marinespecies.org/aphia.php?p=taxdetails&id=155879

**9. P.7 line 11 – Report all _15N values at the same sig fig throughout the study, e.g., 0.1‰ and 0.2‰˙**

We have changed all $\delta^{15}N$ values to one decimal place.

**10. P.10 line 18 – Report TEF as 2.2‰˙**

We have changed the TEF to 2.2 ± 0.3 ‰.

**11. P.10 line 19 – Sig fig of -2‰**

We have changed this to indicate that we used a range of values:

"$\delta^{15}N_{diazo}$ is the isotopic signature of diazotrophs, for which we used a range of -1 to -2 ‰ (Montoya et al., 2002);"

**12. P.11 line 29 – P.12 line 1 – What do you mean? N2 fixation in lagoon lower than mesocosm? Clarify.**

This is clarified in the text as follows:

"N$_2$ fixation rates measured in the lagoon waters were significantly (p<0.05) lower than rates measured in the mesocoms and remained relatively consistent over the 23 days of the experiment (9.2±4.7 nmol N L$^{-1}$ d$^{-1}$)."

*13. P.12 line 2 – What did not differ?*

See response to Question 12.

*14. P.13 line 7 - Do you mean cyclopoid?*

No, we mean poecilostomatoid.

*15. P.13 line 24 – Sig figs. 16. P.16 line 26 – Do you mean $\delta^{15}N$ values of zooplankton?*

Correct, we have amended this to $\delta^{15}N$ values.

**References:**

Gifford, D.J., Caron, D.A., 2000. 5 - Sampling, preservation, enumeration and biomass of marine protozooplankton A2 - Harris, Roger, in: Wiebe, P., Lenz, J., Skjoldal, H.R., Huntley, M. (Eds.), ICES Zooplankton Methodology Manual. Academic Press, London, pp. 193-221.

Hunt, B.P.V., Allain, V., Menkes, C., Lorrain, A., Graham, B., Rodier, M., Pagano, M., Carlotti, F., 2015. A coupled stable isotope-size spectrum approach to understanding pelagic food-web dynamics: A case study from the southwest sub-tropical Pacific. Deep Sea Research Part II: Topical Studies in Oceanography 113, 208-224.

Karl, D., Letelier, R., Tupas, L., Dore, J., Christian, J., Hebel, D., 1997. The role of nitrogen fixation in biogeochemical cycling in the subtropical North Pacific Ocean. Nature 388, 533.

Le Borgne, R., 1987. Biological Production: Benthos and Plankton, in: Longhurst, A.R., Pauly, D. (Eds.), Ecology of Tropical Oceans. Academic Press, San Diego, pp. 106-144.

Le Borgne, R., Rodier, M., Le Bouteiller, A., Kulbicki, M., 1997. Plankton biomass and production in an open atoll lagoon: Uvea, New Caledonia. Journal of Experimental Marine Biology and Ecology 212, 187.

McCutchan, J.H., Lewis, W.M., Kendall, C., McGrath, C.C., 2003. Variation in trophic shift for stable isotope ratios of carbon, nitrogen, and sulfur. Oikos 102, 378-390.

Montoya, J.P., Carpenter, E.J., Capone, D.G., 2002. Nitrogen fixation and nitrogen isotope abundances in zooplankton of the oligotrophic North Atlantic. Limnology and Oceanography 47, 1617-1628.

Sigman, D.M., Altabet, M.A., McCorkle, D.C., Francois, R., Fischer, G., 1999. The $\delta15N$ of nitrate in the southern ocean: Consumption of nitrate in surface waters. Global Biogeochemical Cycles 13, 1149-1166.

Sigman, D.M., Altabet, M.A., Michener, R., McCorkle, D.C., Fry, B., Holmes, R.M., 1997. Natural abundance-level measurement of the nitrogen isotopic composition of oceanic nitrate: an adaptation of the ammonia diffusion method. Marine Chemistry 57, 227-242.

Vanderklift, M.A., Ponsard, S., 2003. Sources of variation in consumer-diet $\delta^{15}N$ enrichment: a meta-analysis. Oecologia V136, 169-182.